# GRASS🌱: Scalable Data Attribution with Gradient Sparsification and Sparse Projection

**Pingbang Hu**[1] **Joseph Melkonian**[2] **Weijing Tang**[3] **Han Zhao**[1] **Jiaqi W. Ma**[1]
[1]University of Illinois Urbana-Champaign    [2]Womp Labs    [3]Carnegie Mellon University
{pbb,hanzhao,jiaqima}@illinois.edu    joe@womplabs.ai    weijingt@andrew.cmu.edu

## Abstract

Gradient-based data attribution methods, such as influence functions, are critical for understanding the impact of individual training samples without requiring repeated model retraining. However, their scalability is often limited by the high computational and memory costs associated with per-sample gradient computation. In this work, we propose GRASS, a novel gradient compression algorithm and its variants FACTGRASS for linear layers specifically, that explicitly leverage the inherent sparsity of per-sample gradients to achieve sub-linear space and time complexity. Extensive experiments demonstrate the effectiveness of our approach, achieving substantial speedups while preserving data influence fidelity. In particular, FACTGRASS achieves up to $165\%$ faster throughput on billion-scale models compared to the previous state-of-the-art baselines.[†]

## 1 Introduction

Data attribution [Deng et al., 2025] aims to measure the impact of individual training samples on a machine learning model and has been widely applied to data-centric problems in modern AI, such as data curation [Koh and Liang, 2017], fact tracing [Lin et al., 2024], and data compensation [Deng et al., 2024b]. There are two major categories of data attribution methods, *gradient-based* and *retraining-based* [Hammoudeh and Lowd, 2024]. The former category, such as influence functions [Koh and Liang, 2017] and its variants, has gained increasing popularity in large-scale applications as it does not require costly model retraining. One common feature of gradient-based methods is their reliance on the *per-sample gradient*—the gradient of the loss with respect to model parameters for each individual data point—to capture the local sensitivity of the model with respect to each training sample, providing a fine-grained understanding of data influence.

However, gradient-based methods still face significant scalability challenges for very large models, such as large language models (LLMs). Specifically, computing and storing per-sample gradients for a model with $n$ training samples and $p$ parameters requires $O(np)$ memory and compute, creating a severe bottleneck for large-scale models. To address this, recent work has explored compressing these high-dimensional gradients into lower-dimensional representations, reducing memory requirements to $O(nk)$ for a target compression dimension $k \ll p$ [Wojnowicz et al., 2016, Park et al., 2023, Choe et al., 2024]. However, this compression often introduces additional computational overhead, as the most common approach—random matrix projection with the Johnson-Lindenstrauss (JL) guarantee—requires dense matrix multiplications, resulting in an overall time complexity of $O(nkp)$.

To overcome these limitations, more specialized approaches have been proposed. For example, the fast Johnson-Lindenstrauss transform (FJLT) used in TRAK's official implementation [Park et al., 2023] exploits structured random matrices, reducing the projection time to $O((p+k)\log p)$ per sample. Alternatively, recent work by Choe et al. [2024] proposed LOGRA that leverages the

---

[†]Our code is publicly available at `https://github.com/TRAIS-Lab/GraSS`.

Kronecker product structure of gradients in linear layers, reducing projection time to $O(\sqrt{pk})$ for each data via a factorized matrix approach. However, these methods are often designed for general inputs and do not fully exploit the unique *sparsity structures* present in per-sample gradients.

In this paper, we push the boundaries of these state-of-the-art (SOTA) gradient compression methods by proposing a novel, two-stage gradient compression algorithm called GRASS (**Gra**dient **S**parsification and **S**parse-projection), that achieves *sub-linear* space and time complexity by explicitly leveraging the inherent sparsity of per-sample gradients. Our contributions are as follows:

1. We identify two critical sparsity properties in per-sample gradients and leverage these to develop a gradient compression algorithm, GRASS, that reduces both space and time complexity from $O(pk)$ to $O(k')$, where $k'$ is a tunable hyperparameter in the range $[k, p]$.

2. We further derive a practically efficient variant of GRASS for linear layers, FACTGRASS (**Fact**orized GRASS), which exploits the gradient structures for linear layers, similarly achieving a time and space complexity of $O(k')$ **without** the need to ever materialize the full gradients.

3. Through extensive experiments, we demonstrate that our approach achieves several orders of magnitude speedup compared to previous SOTA while maintaining competitive performance on standard evaluation metrics. In particular, on billion-scale language models and datasets, FACTGRASS is up to $165\%$ faster in terms of compression throughput compared to LOGRA.

## 2 Preliminary

We begin by introducing the influence function and its practical implementation as our running example of gradient-based data attribution methods. We then demonstrate how random projection can be integrated with the influence function. We note that the proposed compression methods naturally extend to other gradient-based approaches that share similar low-level computations. Examples include TRAK [Park et al., 2023], SGD-Influence [Hara et al., 2019], and Data Value Embedding [Wang et al., 2025]. A more detailed discussion of related work is provided in Section A.1.

### 2.1 Influence function

Given a dataset $D = \{z_i \in \mathbb{R}^d\}_{i=1}^n$, consider a model parametrized by $\hat{\theta} \in \mathbb{R}^p$ that this trained on the dataset $D$ with a loss function $\ell \colon \mathbb{R}^d \times \mathbb{R}^p \to \mathbb{R}$ via empirical risk minimization: $\hat{\theta} = \arg\min_{\theta \in \mathbb{R}^p} \frac{1}{n} \sum_{i=1}^n \ell(z_i; \theta)$. Under this setup, the *influence function* [Koh and Liang, 2017] can be theoretically derived; essentially, it gives an estimation on every training data $z_i$'s "influence" $\mathcal{I}(z_i, z_{\text{test}})$ of the test loss $\ell(z_{\text{test}}; \hat{\theta})$ of a given test data $z_{\text{test}}$ when $z_i$ is removed from $D$ as

$$\mathcal{I}(z_i, z_{\text{test}}) \coloneqq \nabla_\theta \ell(z_{\text{test}}; \hat{\theta})^\top H_{\hat{\theta}}^{-1} \nabla_\theta \ell(z_i; \hat{\theta}),$$

where $H_{\hat{\theta}} = \frac{1}{n} \sum_{i=1}^n \nabla_\theta^2 \ell(z_i, \hat{\theta})$ is the empirical Hessian. As $H_{\hat{\theta}} \in \mathbb{R}^{p \times p}$ and computing it requires higher-order differentiation for every training data, several approximation algorithms aim to mitigate this. One famous approximation is the Fisher information matrix (FIM) [Fisher, 1922] approximation $H_{\hat{\theta}} \approx \mathbb{E}_z[\nabla_\theta \ell(z; \hat{\theta}) \nabla_\theta \ell(z; \hat{\theta})^\top]$, which is exact for model trained with the negative log-likelihood objective. Since FIM only involves the first-order gradient, which is also needed in the other parts of the calculation of influence $\mathcal{I}$, and hence is a popular and efficient approximation. Given this, people realize that an efficient way to compute the influence function is to divide the computation into two stages [Lin et al., 2024, Choe et al., 2024]:

1. **Cache stage**: 1.) compute all *per-sample gradients* $g_i \coloneqq g_{z_i} \coloneqq \nabla_\theta \ell(z_i; \hat{\theta})$, 2.) construct the FIM $F_{\hat{\theta}} \coloneqq \frac{1}{n} \sum_{i=1}^n g_i g_i^\top$, 3.) perform *inverse FIM-vector-product* (iFVP) via $\widetilde{g}_i \coloneqq F_{\hat{\theta}}^{-1} g_i$ for all $z_i$'s.

2. **Attribute stage**: For a query data $z_{\text{test}}$, 1.) compute its per-sample gradient $g_{\text{test}} \coloneqq \nabla_\theta \ell(z_{\text{test}}; \hat{\theta})$, 2.) compute all-pair-inner-product between $g_{\text{test}}$ and $\{\widetilde{g}_i\}_{i=1}^n$ as $\mathcal{I}(z_i, z_{\text{test}}) = \langle g_{\text{test}}, \widetilde{g}_i \rangle$ for all $z_i$'s.

The bottleneck of this pipeline is the cache stage, since the problem for the attribute stage is the well-studied *vector inner product search*, where numerous optimization techniques have been studied in the vector database community. On the other hand, iFVP remains a challenging task due to the matrix inversion of quadratic model size, where in most cases, even materializing FIM is infeasible.

## 2.2 Random projection

Despite multiple attempts to accelerate influence function from various angles, one of the most naive and simple strategies, RANDOM [Wojnowicz et al., 2016, Schioppa et al., 2022, Park et al., 2023], remains practically relevant and achieves SOTA attribution results. RANDOM leverages sketching (random projection) techniques by replacing each per-sample gradient $g_i \in \mathbb{R}^p$ with $\hat{g}_i := Pg_i \in \mathbb{R}^k$, where $P \in \mathbb{R}^{k \times p}$ is a random projection matrix for some $k \ll p$. This subsequently leads to the *projected FIM* approximation $\hat{F}_{\hat{\theta}} := PF_{\hat{\theta}}P^\top = \mathbb{E}_z[(\hat{g}_z \cdot (\hat{g}_z)^\top)] \in \mathbb{R}^{k \times k}$, i.e., a restriction of $F_{\hat{\theta}}$ to the subspace spanned by the columns of $P$. The theoretical merits of RANDOM largely come from the well-known Johnson-Lindenstrauss lemma [Johnson, 1984], which states that for $P$ drawn appropriately, e.g., $P_{ij} \overset{\text{i.i.d.}}{\sim} \mathcal{N}(0,1)$ or $\mathcal{U}(\{\pm 1\})$ for all $i, j$,[1] with high probability, the pair-wise distance $\|g_i - g_j\|$ between any two $g_i$ and $g_j$ will be preserved up to $1 \pm \epsilon$ factor after the projection, whenever $k = O(\epsilon^{-2} \log n)$. While this does not fully justify whether the inner product between a projected per-test-sample gradient $\hat{g}_{\text{test}}$ and the "conditioned" projected per-train-sample gradient $\widehat{\widetilde{g}}_i := (\hat{F}_{\hat{\theta}})^{-1}\hat{g}_i$ will be preserved (see Section A.2 for an in-depth discussion), RANDOM remains to be one of the strongest baselines to date and is practically appealing due to its simplicity.[2]

Computational-wise, RANDOM accelerates iFVP significantly as the matrix inversion complexity scales down from $O(p^2)$ to $O(k^2)$. In terms of the projection overhead, the matrix-based projection method requires $O(kp)$ overhead per projection. TRAK [Park et al., 2023] leverages the *fast Johnson-Lindenstrauss transform* (FJLT) [Ailon and Chazelle, 2009, Fandina et al., 2023] that has a similar theoretical guarantee as the random matrix-based projection to achieve a speed up of $O((p+k)\log p)$. Another line of work by Choe et al. [2024] called LOGRA exploits the gradient structure of linear layers and factorizes the projection accordingly, reducing the problem size quadratically. With suitable hyper-parameter choice, the computational complexity goes down from $O(k_l p_l)$ to $O(\sqrt{k_l p_l})$, where $k_l$ and $p_l$ now refer to the projection dimension and number of model parameters of one ($l^{\text{th}}$) linear layer. This sets the SOTA efficiency and attribution quality to date.

# 3  GRASS: Gradient Sparsification and Sparse projection

In this section, we first explore two key sparsity properties in per-sample gradients (Sections 3.1 and 3.2) and propose efficient compression methods for each. Combining them, we present GRASS and its variant FACTGRASS (Section 3.3), which beat the previous SOTA data attribution algorithms.

## 3.1  Per-sample gradient sparsity

Modern deep learning models often induce highly sparse per-sample gradients, especially when using popular activation functions like ReLU [Nair and Hinton, 2010]. To see this, consider the gradient of the first read-in linear layer with weight $W \in \mathbb{R}^{d^{\text{out}} \times d^{\text{in}}}$ with $d^{\text{in}} = d$ and ReLU activations. Then given a sample $z \in \mathbb{R}^d$, the output is $h = \text{ReLU}(Wz)$. Since $\text{ReLU}(x) = \max(0, x)$ sets all negative pre-activations to zero, naturally creating sparse activations. This sparsity propagates to the gradient computations via the chain rule, resulting in gradients with numerous zero entries. This is not unique to ReLU and extends to many other activation functions that exhibit similar behavior.

**Remark 3.1.** *Such sparsity is unique to* per-sample *gradients: for mini-batch gradients $\sum_{i \in B} g_i / |B|$, the sparsity pattern differs for individual $g_i$ and will be destroyed when adding together.*

Given the inherently sparse nature of these gradients, it is natural to consider other compression methods that can effectively exploit this input sparsity. Traditional dense random projection methods struggle to leverage sparse inputs without incurring significant overhead. Although FJLT is often more efficient, its algorithmic structure also prevents it from effectively exploiting sparsity patterns.

**Sparse Johnson-Lindenstrauss Transform.** A natural candidate for efficient gradient projection is the *sparse Johnson-Lindenstrauss transform* (SJLT) [Dasgupta et al., 2010, Kane and Nelson, 2014], which significantly reduces the computational cost by sparsifying the projection matrix. To understand SJLT, it is useful to revisit the standard matrix-based projection approach, which relies

---

[1]We omit the normalization factor (in this case, $1/\sqrt{k}$) to keep the presentation clean.

[2]We follow the same notational convention in the rest of the paper: $\hat{\cdot}$ denotes compression, $\widetilde{\cdot}$ denotes (FIM) precondition, and $\widehat{\widetilde{\cdot}}$ denotes the compression and then (compressed FIM) precondition.

on matrix-vector multiplication. Given a projection matrix $P \in \mathbb{R}^{k \times p}$ and an input vector $g \in \mathbb{R}^p$ to be projected, the product $Pg$ can be computed as $\hat{g} = Pg = \sum_{j=1}^{p} g(j) P_{:j}$, where the $j^{\text{th}}$ term represents the $j^{\text{th}}$ column $P_{:j}$ of $P$ scaled by the $j^{\text{th}}$ entry $g(j)$ of $g$. In the case of a dense Rademacher projection matrix (entries being $\pm 1$), this requires $O(pk)$ computation for both constructing $g(j)P_{:j}$ and summing them. This dense projection process is illustrated in Figure 1.

$$P \quad \times \quad g \ = \ P_{:1} \times g(1) \ + \cdots + \ P_{:p} \times g(p) = \ \hat{g} \qquad\qquad P \quad \times \quad g \ = \ P_{:1} \times g(1) \ + \cdots + \ P_{:p} \times g(p) = \ \hat{g}$$

Figure 1: Dense Rademacher projection.       Figure 2: Sparse Rademacher projection ($s = 1$).

From this perspective, the SJLT arises naturally: by "zeroing" out the projection matrix $P$, we significantly reduce the required computation, as illustrated in Figure 2. Specifically, Dasgupta et al. [2010] and Kane and Nelson [2014] demonstrated that retaining only $s = o_\epsilon(k)$ out of the $k$ possible non-zero entries for each column of the projection matrix still preserves the essential properties required by the Johnson-Lindenstrauss lemma. This approach, which we denote as $\text{SJLT}_k(\cdot)$, reduces both the time and space complexity to $O(ps)$, where $s = o_\epsilon(k)$ is much smaller than $k$.

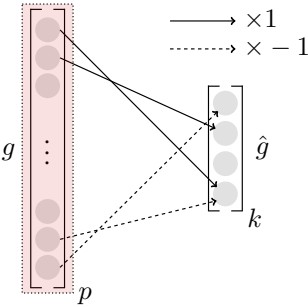

An equivalent way to view the computation of SJLT is shown in Figure 3, where we initialize $\hat{g}$ to be a zero vector and sequentially scan through $g$, with each $g(j)$ for $j \in [p]$ chooses $s$ many random $j' \in [k]$ to either add on or subtract from the corresponding $\hat{g}(j')$.

Figure 3: SJLT with $s = 1$.

It is immediate that if the input is **itself sparse**, the complexity can be further reduced. Specifically, for a dense matrix projection, the complexity becomes $O(k \, \text{nnz}(g))$, and for SJLT, this drops to $O(s \, \text{nnz}(g))$, where $\text{nnz}(g) \coloneqq \|g\|_0$ denotes the number of non-zero entries in $g$. We highlight $g$ in red in Figure 3 to signify that the computational complexity of SJLT scales with the size of $g$.

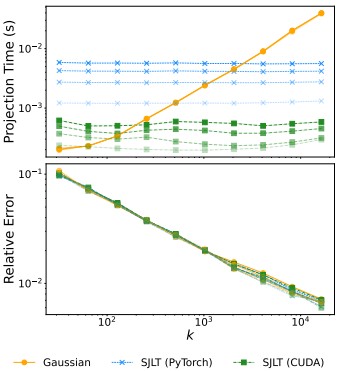

Figure 4: Benchmark of different projection methods with $p = 131{,}072$ under several sparsity levels (distinguished by opacity). Relative error is w.r.t. pair-wise distance preservation.

**Implementation of SJLT.** Despite these theoretical advantages, practical implementations of SJLT face critical performance challenges, such as thread contention and irregular memory access patterns. While the latter overhead is due to the nature of SJLT, the former occurs because multiple threads may attempt to write to the same entry in the output vector $\hat{g}$, causing race conditions that degrade performance, especially when the target dimension $k$ is small. These are especially critical when implementing in general-purpose libraries like `PyTorch`. Moreover, the default matrix multiplication algorithms in `PyTorch` are highly hardware-optimized (e.g., cache-friendly memory layouts and fused multiply-accumulate instructions), in practice often outperforming any similar multiplication algorithms when the problem size is small.

To address these issues, we developed a *SJLT CUDA kernel*[3] that optimizes the memory access patterns and minimizes thread contention to better exploit the underlying hardware capabilities. This kernel significantly reduces the overhead compared to its `PyTorch` implementation counterpart, resulting in substantial performance gains. As shown in Figure 4, for SJLT with $s = 1$, our CUDA implementation outperforms the highly optimized dense matrix multiplications for small projection problem sizes, while retaining the speedup of SJLT w.r.t. input sparsity. In contrast, dense Gaussian projections exhibit a clear dependency on $k$ while neglecting the input sparsity, making them less efficient in such cases. In practice, we set $s = 1$ to optimize for speed while enjoying a strong empirical guarantee for small relative error, as seen in Figure 4.

---

To summarize, the complexity of SJLT 1.) scales with the input sparsity, and 2.) is independent of the target dimension $k$, both of which are critical for efficient gradient compression. Specifically, 1.) per-sample gradients are naturally sparse, and 2.) larger $k$ generally improves the fidelity of data attribution, which is more desirable. These make SJLT a natural fit for gradient compression.

## 3.2 Effective parameter sparsity

While SJLT effectively reduces the computational overhead and takes advantage of the sparsity structure of the input, it still scales linearly with the potentially large input dimensionality $p$. We now explore a more aggressive compression that achieves *sub-linear* complexity by directly exploiting the inherent *effective parameter sparsity* in neural networks. We term this approach as *sparsification*.

**Random Mask.** Modern deep learning models often exhibit a high degree of parameter redundancy, where only a small fraction of the weights significantly contribute to the model's final performance [Han et al., 2016, Frankle and Carbin, 2019]. In a similar vein, the distributed training community has observed that the majority of gradient updates are redundant, allowing for substantial compression without a significant impact on model accuracy [Lin et al., 2018, Aji and Heafield, 2017]. This suggests that many parameters can be safely ignored without substantial loss in accuracy. Inspired by this observation, a simple yet surprisingly effective sparsification algorithm, *Random Mask*, randomly selects a small subset from the $p$ input dimensions to form a compressed representation.

Formally, the Random Mask $(\text{RM}_k(\cdot))$ involves selecting a random subset of $k$ dimensions from the original $p$-dimensional gradient vector, effectively extracting a length-$k$ sub-vector. This can also be viewed as a random projection onto the standard basis of a randomly chosen $k$-dimensional Euclidean subspace, i.e., $\hat{g} = Mg$ where $M \in \mathbb{R}^{k \times p}$ is a sparse binary selection matrix with exactly one (non-repetitive) non-zero entry per row, corresponding to the randomly chosen dimensions. This is illustrated in Figure 5.

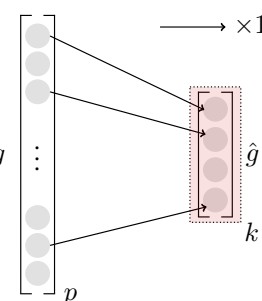

Figure 5: Mask.

At first glance, $\text{RM}_k$ may seem overly aggressive, as it discards a substantial amount of information. However, empirical evidence suggests that this method can still yield non-trivial attribution performance, especially when the underlying gradient distribution is sparse or when the model is over-parameterized. Moreover, the extreme simplicity of this approach makes it highly efficient, with a computational cost of just $O(k)$, achieving a *sub-linear* complexity w.r.t. $p$. We highlight $\hat{g}$ in red in Figure 5 to signify that the computational complexity of Random Mask scales with the size of $\hat{g}$.

**Selective Mask.** Building on the idea of Random Mask, we introduce a more structured approach, *Selective Mask* $(\text{SM}_k(\cdot))$, which aims to selectively retain the most important parameters based on a simple, yet effective, data-driven optimization. Inspired by recent work on identifying influential model parameters [He et al., 2025], Selective Mask introduces a small but meaningful optimization overhead to improve the fidelity of the compressed representation. Formally, given a training set $\{z_i\}_{i=1}^n$, we define the selective masking problem as the following *unconstrained* optimization task:

$$S^* = \arg\max_{S \in \mathbb{R}^p} \mathbb{E}_{z_{\text{test}}} \left[ \text{corr}\left( (\langle g_i, g_{z_{\text{test}}} \rangle)_{i=1}^n, (\langle \hat{g}_i, \hat{g}_{z_{\text{test}}} \rangle)_{i=1}^n \right) \right] - \lambda \|\sigma(S)\|_1, \tag{1}$$

where $\hat{g}_i = \sigma(S) \odot g_i \in \mathbb{R}^p$ is the (soft-)masked $g_i$, $\odot$ denotes the element-wise product, and $\sigma(\cdot) \in (0,1)$ is the sigmoid function. The first term of the objective encourages the average correlation between the original and masked gradients' GRADDOT attribution scores [Charpiat et al., 2019], a widely used and computationally efficient approximation of the influence function. The second term, an $\ell_1$ regularization penalty, promotes sparsity by pushing $\sigma(S)$ towards a binary mask.

Once the optimal $S^*$ is obtained after solving Eq. (1), the final binary mask $M \in \{0,1\}^{k \times p}$ can be extracted by thresholding the sigmoid outputs with $k = \sum_{j=1}^p \mathbb{1}_{\sigma(S^*)_j \geq 0.5}$ is the number of selected dimensions. Formally, one can obtain the explicit mask matrix $M$ via solving $\langle M_{:j}, 1_k \rangle = \mathbb{1}_{\sigma(S^*)_j \geq 0.5}$, but the actual implementation is simply an index extraction for all $j$ for $\mathbb{1}_{\sigma(S^*)_j \geq 0.5}$.

This formulation avoids the exponential complexity of directly optimizing over discrete binary masks, as the continuous nature of $S$ allows for efficient, first-order gradient-based optimization. While it incurs a one-time overhead for solving Eq. (1), this method provides a more principled approach to

mask selection by directly targeting a widely used surrogate data attribution score, making it a natural extension of Random Mask. We use $\text{MASK}_k$ to refer to either $\text{RM}_k$ or $\text{SM}_k$ for convenience.

### 3.3 GRASS & FACTGRASS: Multi-stage compression

We now formally introduce GRASS and FACTGRASS, an integration of the proposed approaches by combining the sparse projection (Section 3.1) and also sparsification techniques (Section 3.2).

#### 3.3.1 GRASS: Sparsify first, sparse projection next

Recall that the time complexity of SJLT with $s = 1$ is $O(p)$ where $p$ is the input dimension, while for both sparsification techniques are $O(k)$ where $k$ is the target sparsification dimension. A natural idea is to employ a two-stage compression: Given an input $g$ and target compression dimension $k \ll p$,

1. **Sparsification**: sparsify the input $g$ to a sub-vector $g'$ of dimension $k'$ with $k < k' \ll p$.

2. **Sparse projection**: then apply SJLT to $g'$ to get the compression $\hat{g}$ with target dimension $k$.

We term this simple per-sample gradient compression method **Gradient Sparsification and Sparse projection** (GRASS), which is illustrated in Figure 6. This leads to a *sub-linear* time complexity $O(k' + k') = O(k')$ to the input dimension $p$, since the runtime of SJLT depends only on its input dimension, which is now sparsified to $k'$ from $p$. In the extreme cases when $k' = p$, GRASS reduces to vanilla SJLT; while when $k' = k$, GRASS reduces to sparsification. Notation-wise, we write $\text{SJLT}_k \circ \text{MASK}_{k'}$.

Intuitively, $k'$ as a hyperparameter balances the computational complexity and attribution performance: after selecting $k'$ coordinates of $g$ to form $g'$, subsequent application of SJLT will compress $g'$ down to the target dimension without losing the pair-wise distance information.

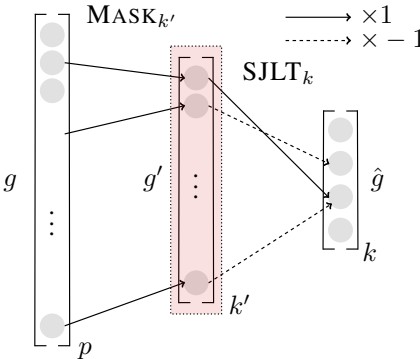

Figure 6: GRASS.

#### 3.3.2 FACTGRASS: Exploiting layer-wise gradient factorization structure

In addition to FIM approximation, influence function on large-scale models often leverages the *layer-wise independence* assumption by approximating FIM as a block-diagonal matrix, ignoring parameter interactions across layers. Specifically, this approach decomposes the FIM as $\text{diag}\{F_{\hat{\theta}_1}, \cdots, F_{\hat{\theta}_L}\}$ for an $L$-layer neural network, where each block $F_{\hat{\theta}_l}$ corresponds to the $l^{\text{th}}$ layer's parameters $\hat{\theta}_l$, defined as $F_{\hat{\theta}_l} = \mathbb{E}_z[\nabla_{\theta_l}\ell(z; \hat{\theta})\nabla_{\theta_l}\ell(z; \hat{\theta})^\top]$. By writing $g_{i,l} \coloneqq \nabla_{\theta_l}\ell(z_i; \hat{\theta})$, iFVP computation can now be done layer-wise as $\widetilde{g}_{i,l} = F_{\hat{\theta}_l}^{-1} g_{i,l}$. Furthermore, coupling this trick with gradient compression, i.e., consider compressing $g_{i,l}$ to $\hat{g}_{i,l}$, which subsequently forms $\hat{F}_{\hat{\theta}_l}$, we now compute $\widetilde{\hat{g}}_{i,l} \coloneqq \hat{F}_{\hat{\theta}_l}^{-1} \hat{g}_{i,l}$.

However, this renders a critical challenge for GRASS to demonstrate practical speedup via a direct application of each of these layer-wise compression sub-problems, since SJLT suffers from small problem sizes (Section 3.1). Specifically, if each layer has roughly the same number of parameters, the compression problem size is reduced from $p \times k$ to $p/L \times k/L$ each if the compression dimension is allocated uniformly. Moreover, recent techniques such as LOGRA [Choe et al., 2024] further reduce the compression problem size of each layer-wise compression via *gradient factorization*, making it even more difficult to integrate GRASS with these SOTA methods to achieve a further speedup.

This motivates the need for a specialized adaptation of GRASS that can effectively exploit a similar gradient factorization structure. To this end, we propose FACTGRASS, which explicitly incorporates this factorization structure to achieve even greater efficiency in gradient compression.

**Recap on LOGRA.** To motivate and understand the practical difficulties we must avoid, we first introduce LOGRA. Formally, LOGRA exploits the *factorized structure* of linear layer's gradients. For full generality, consider a sequential input $z_i \in \mathbb{R}^{d \times T}$ of length $T$ to the model and the corresponding input $z_{i,l}^{\text{in}} \in \mathbb{R}^{d_l^{\text{in}} \times T}$ and output (pre-activations) $z_{i,l}^{\text{out}} \in \mathbb{R}^{d_l^{\text{out}} \times T}$ of the $l^{\text{th}}$ linear layer with a weight

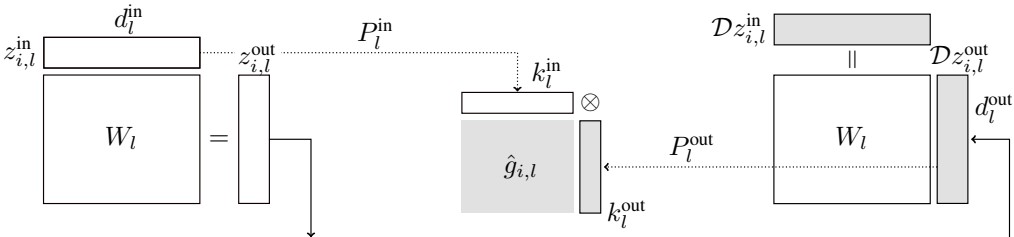

Figure 7: LoGra for one linear layer. Note that the application of $\text{vec}(\cdot)$ on $\hat{g}_{i,l}$ is omitted.

matrix $W_l \in \mathbb{R}^{d_l^{\text{out}} \times d_l^{\text{in}}}$ such that $z_{i,l}^{\text{out}} = W_l z_{i,l}^{\text{in}}$. Then the gradient of the $l^{\text{th}}$ linear layer is given by

$$\frac{\partial \ell(z_i; \hat{\theta})}{\partial W_l} = \frac{\partial \ell(z_i; \hat{\theta})}{\partial z_{i,l}^{\text{out}}} \frac{\partial z_{i,l}^{\text{out}}}{\partial W_l} = \frac{\partial \ell(z_i; \hat{\theta})}{\partial z_{i,l}^{\text{out}}} z_{i,l}^{\text{in}\top} \iff \text{vec}(\mathcal{D} W_l) = \sum_{t=1}^{T} (z_{i,l}^{\text{in}})_{:t} \otimes \mathcal{D}(z_{i,l}^{\text{out}})_{:t}, \quad (2)$$

where we write $\nabla_v \ell(z_i; \hat{\theta})$ as $\mathcal{D}v$ for any $v$. LoGra then leverages this Kronecker-product structure of $\text{vec}(\mathcal{D} W_l)$ by assuming the projection matrix $P_l$ for this layer has a factorized structure, i.e.,

$$P_l \text{vec}(\mathcal{D} W_l) := (P_l^{\text{in}} \otimes P_l^{\text{out}}) \text{vec}(\mathcal{D} W_l) = \sum_{t=1}^{T} (P_l^{\text{in}} (z_{i,l}^{\text{in}})_{:t}) \otimes (P_l^{\text{out}} \mathcal{D}(z_{i,l}^{\text{out}})_{:t}), \quad (3)$$

where $P_l^{\text{in}} \in \mathbb{R}^{k_l^{\text{in}} \times d_l^{\text{in}}}$, $P_l^{\text{out}} \in \mathbb{R}^{k_l^{\text{out}} \times d_l^{\text{out}}}$, and $P_l = P_l^{\text{in}} \otimes P_l^{\text{out}} \in \mathbb{R}^{(k_l^{\text{in}} k_l^{\text{out}}) \times (d_l^{\text{in}} d_l^{\text{out}})}$. Naturally, we let $k_l := k_l^{\text{in}} \times k_l^{\text{out}}$ and $p_l := d_l^{\text{in}} \times d_l^{\text{out}}$ to be the compression dimension and number of parameters for the $l^{\text{th}}$ linear layer, respectively. Hence, $P_l \in \mathbb{R}^{k_l \times p_l}$ as we expect, with $p = \sum_{l=1}^{L} p_l$ and $k = \sum_{l=1}^{L} k_l$. The computation of $P_l \text{vec}(\mathcal{D} W_l)$ of LoGra is illustrated in Figure 7, where:

- forward pass on $z_i$ and backward pass on $\ell(z_i; \hat{\theta})$ give $z_{i,l}^{\text{in}}$ and $\mathcal{D} z_{i,l}^{\text{out}}$ for each of the $l^{\text{th}}$ linear layer;

- only two smaller projection problems of size $k_l^{\text{in}} \times d_l^{\text{in}}$ and $k_l^{\text{out}} \times d_l^{\text{out}}$ are needed for each sequential index, instead of to project the entire gradient, which is of size $(k_l^{\text{in}} k_l^{\text{out}}) \times (d_l^{\text{in}} d_l^{\text{out}}) = k_l \times p_l$;

- in addition, the actual gradient of the layer is *never* materialized (which will require computing Eq.(2), $O(Tp_l)$), only the projected gradient is materialized at the end (which takes $O(Tk_l)$).

Notation-wise, as $P_l^{\text{in}}$ and $P_l^{\text{out}}$ are default to Gaussian projection [Choe et al., 2024], we write LoGra as $\text{Gauss}_{k_l^{\text{in}} \otimes k_l^{\text{out}}}$, where $\otimes$ indicates that the projection is done in a factorized manner. We see that overall, assuming $d_l^{\text{in}} \approx d_l^{\text{out}} \approx \sqrt{p_l}$, choosing $k_l^{\text{in}} \approx k_l^{\text{out}} \approx \sqrt{k_l}$ results in a speedup from $O(k_l p_l)$ to $O(\sqrt{k_l p_l})$ per projection (i.e., per input and per sequential index) for the $l^{\text{th}}$ layer.

**Bottlenecks of integrating GraSS with LoGra.** One can change the (dense) Gaussian projection used in LoGra to other compression methods, e.g., SJLT, Mask, or GraSS, resulting in $\text{Mask}_{k_l^{\text{in}} \otimes k_l^{\text{out}}}$, $\text{SJLT}_{k_l^{\text{in}} \otimes k_l^{\text{out}}}$, and $\text{GraSS}_{k_l^{\text{in}} \otimes k_l^{\text{out}}}$, respectively. However, a trivial integration with GraSS will **not** lead to a practical speed up compared to LoGra since each compression problem size is now reduced, making GraSS slower than Gaussian projection due to the practical implementation overhead of one of its algorithmic components, SJLT, as shown in Figure 4. This small projection problem size regime makes a direct integration of GraSS with LoGra challenging.

A natural idea to mitigate this issue is to apply SJLT to a **moderate** dimension by **not** factorizing the projection: by first constructing the gradients of the layer via Eq.(2), we can perform GraSS on a much larger problem size ($p_l \times k_l$) rather than the two smaller problems (roughly $\sqrt{p_l} \times \sqrt{k_l}$). However, this results in another bottleneck: materializing the gradients explicitly blows up the space and time complexity to $O(p_l)$, which is slower than LoGra, defeating the whole purpose.

**Factorized GraSS 🌿.** To bypass the bottlenecks, we propose *Factorized* GraSS (FactGraSS), a variant of GraSS that exploits the Kronecker-product structure. Specifically, given a target compression dimension $k_l = k_l^{\text{in}} \times k_l^{\text{out}} \ll p_l$ for the $l^{\text{th}}$ layer, after a forward pass on $z_i$ and a backward pass on $\ell(z_i; \hat{\theta})$ to get $z_{i,l}^{\text{in}}$ and $\mathcal{D} z_{i,l}^{\text{out}}$, FactGraSS operates in three stages (Figure 8):

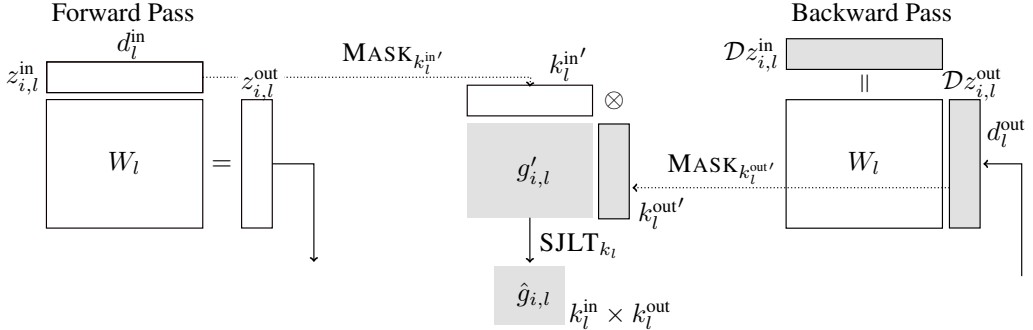

Figure 8: FACTGRASS for one linear layer. Note that the output of SJLT is a vector in practice.

1. **Sparsification**: sparsify both $z_{i,l}^{\text{in}}$ and $\mathcal{D}z_{i,l}^{\text{out}}$ to an intermediate dimension $k_l^{\text{in}'}$ and $k_l^{\text{out}'}$, where $k_l^{\text{in}} \le k_l^{\text{in}'} \ll d_l^{\text{in}}$ and $k_l^{\text{out}} \le k_l^{\text{out}'} \ll d_l^{\text{out}}$, respectively;

2. **Reconstruction**: construct the "sparsified gradient" $g_{i,l}'$ of dimension $k_l' := k_l^{\text{in}'} \times k_l^{\text{out}'}$ via Eq. (3), i.e., Kronecker product between the sparsified $z_{i,l}^{\text{in}}$ and the sparsified $\mathcal{D}z_{i,l}^{\text{out}}$;

3. **Sparse projection**: apply SJLT to $g_{i,l}'$ to get the compressed $\hat{g}_{i,l}$ with target dimension $k_l$.

Intuitively, FACTGRASS resolves the two bottlenecks by 1.) avoid reconstructing the *full gradient* via sparsification, and 2.) avoid small problem size for SJLT via reconstruction. In terms of complexity, sparsification takes $O(k_l^{\text{in}'})$ and $O(k_l^{\text{out}'})$ respectively, and reconstruction takes $O(k_l')$ where $k_l' := k_l^{\text{in}'} \times k_l^{\text{out}'}$ for performing the Kronecker product between two vectors of size $k_l^{\text{in}'}$ and $k_l^{\text{out}'}$, finally sparse projection also takes $O(k_l')$, giving an overall time and space complexity of $O(k_l')$.

Notation-wise, we write FACTGRASS as $\text{SJLT}_{k_l} \circ \text{MASK}_{k_l^{\text{in}'} \otimes k_l^{\text{out}'}}$, where we use $\otimes$ to indicate that the sparsification is done in a factorized manner. The following summarizes the complexity of our proposed methods (GRASS and FACTGRASS), as well as other baselines we have mentioned:

| Compression | General | | | | | Linear Layer | |
| | Sparsification | Sparse Projection | GRASS | Baselines | | FACTGRASS | LOGRA (Baseline) |
| | $\text{MASK}_k$ | $\text{SJLT}_k$ | $\text{SJLT}_k \circ \text{MASK}_{k'}$ | $\text{GAUSS}_k$ | $\text{FJLT}_k$ | $\text{SJLT}_{k_l} \circ \text{MASK}_{k_l^{\text{in}'} \otimes k_l^{\text{out}'}}$ | $\text{GAUSS}_{k_l^{\text{in}} \otimes k_l^{\text{out}}}$ |
|---|---|---|---|---|---|---|---|
| Complexity | $O(k)$ | $O(p)$ | $O(k')$ | $O(pk)$ | $O((p+k)\log p)$ | $O(k_l')$ per-layer | $O(\sqrt{p_l k_l})$ per-layer |

In particular, compared to LOGRA, by writing $k_l' \in [k_l, p_l]$ as $k_l' = ck_l$ for some *blow-up factor* $c \ge 1$, FACTGRASS is theoretically faster than LOGRA if $k_l' \le \sqrt{k_l p_l}$, or equivalently, $c \le \sqrt{p_l / k_l}$. In practice, this is easy to satisfy: for instance, consider a linear layer of size $p_l = 4096 \times 4096$ with $k_l = 64 \times 64$. In this case, for any blow-up factor $c \le 64$, FACTGRASS is faster than LOGRA.

With $k' := \sum_{l=1}^{L} k_l'$, FACTGRASS overall takes $O(k')$ per compression, same as GRASS **without** ever materializing the full gradient. We summarize both GRASS and FACTGRASS in Theorem 3.2:

**Theorem 3.2.** *There is a sub-linear compression algorithm with a complexity of $O(k')$ per sample, where $k < k' \ll p$. Moreover, this extends to linear layers, where gradients are never materialized.*

We conclude by noting that much of the recent gradient-based data attribution literature could benefit from per-sample gradient compression, underscoring the broad applicability of our methods.

## 4 Experiment

In this section, we evaluate the effectiveness of GRASS and FACTGRASS in terms of *accuracy* and *efficiency*. Specifically, in Section 4.1, we first perform the standard counterfactual evaluations to quantitatively study the data valuation accuracy of GRASS and FACTGRASS on small-scale setups. Then, we scale FACTGRASS to a billion-scale model and billion-token dataset, where we investigate the qualitative accuracy and memory/compute efficiency in Section 4.2. Further experimental details, such as hyperparameters and compute resources, can be found in Section B.1.

## 4.1 Quantitative accuracy via counterfactual evaluation

We assess the quantitative accuracy of data attribution algorithms using the widely adopted *linear datamodeling score* (LDS) [Park et al., 2023], a counterfactual evaluation method. While LDS relies on the additivity assumption, which is known to be imperfect [Hu et al., 2024], it remains a valuable evaluation metric for data attribution. All the quantitative experiments are conducted on one NVIDIA A40 GPU with 48 GB memory, and other details can be found in Section B.2.

**GRASS with TRAK.** We apply GRASS on one of the SOTA data attribution algorithms (in terms of attribution quality), TRAK [Park et al., 2023], with the implementation from the dattri library [Deng et al., 2024a]. To validate the effectiveness of our sparsification and sparse projection methods, we conduct an ablation study on a simple 3-layer MLP trained on MNIST [LeCun, 1998]. As shown in Table 1(a), even a standalone Random Mask achieves non-trivial LDS results, while Selective Mask improves the performance further. Additionally, the sparse projection SJLT significantly outperforms baselines like FJLT and Gaussian projection in both efficiency and LDS accuracy.

We evaluate GRASS on more complex models: 1.) ResNet9 [He et al., 2016] with CI-FAR2 [Krizhevsky and Hinton, 2009], and 2.) Music Transformer [Huang et al., 2019] with MAE-STRO [Hawthorne et al., 2019]. Results in Tables 1(b) and 1(c) demonstrate that while sparsification methods are highly efficient, they often fall short in LDS performance. In contrast, sparse projection methods achieve competitive LDS scores but typically incur higher projection costs, though they still outperform the baseline[4] FJLT by a large margin. Notably, GRASS strikes a balance between these extremes, achieving competitive LDS scores at a fraction of the computational cost.

**FACTGRASS with Influence Function on Linear Layers.** We next evaluate FACTGRASS with layer-wise block-diagonal FIM influence functions for linear layers. We consider a small language model, GPT2-small [Radford et al., 2019] fine-tuned on the WikiText dataset [Merity et al., 2016], to enable LDS evaluation. The results are presented in Table 1(d), where $k_l$ indicates the target compression dimension for each linear layer. We further set $k_l^{in} = k_l^{out} = \sqrt{k_l}$ for simplicity. As discussed in Section 3.3.2, replacing the Gaussian projection matrices in LOGRA with SJLT most likely will result in an efficiency degradation, although it achieves a competitive LDS. On the other hand, standalone sparsification achieves competitive LDS results with minimal compression overhead, highlighting its potential as an efficient alternative in overparametrized models. Finally, FACTGRASS not only maintains the LDS performance of SJLT but also significantly improves computational efficiency, achieving up to a $250\%$ speedup over the most efficient SOTA baseline LOGRA.

Table 1: Quantitative evaluation results with different gradient compression methods.

(a) LDS and compression wall-time for MLP with MNIST on TRAK.

|  | $\text{RM}_k$ | | | $\text{SM}_k$ | | | $\text{SJLT}_k$ | | | $\text{FJLT}_k$ | | | $\text{GAUSS}_k$ | | |
|  | Sparsification | | | | | | Sparse Projection | | | Baselines | | | | | |
| $k$ | 2048 | 4096 | 8192 | 2048 | 4096 | 8192 | 2048 | 4096 | 8192 | 2048 | 4096 | 8192 | 2048 | 4096 | 8192 |
|---|---|---|---|---|---|---|---|---|---|---|---|---|---|---|---|
| LDS | 0.3803 | 0.4054 | 0.4318 | 0.3882 | 0.4163 | **0.4373** | **0.4171** | **0.4280** | 0.4357 | 0.4146 | 0.4359 | 0.4347 | 0.4101 | 0.4253 | 0.4346 |
| Time (s) | 0.1517 | 0.1458 | 0.1501 | **0.1354** | **0.1346** | **0.1487** | 0.4919 | 0.5172 | 0.4754 | 0.8997 | 1.4341 | 2.4387 | 3.0806 | 5.5421 | 10.8355 |

(b) LDS and compression wall-time for ResNet9 with CIFAR2 on TRAK.

|  | Sparsification | | | | | | Sparse Projection | | | GRASS | | | | | | Baseline | | |
|  | $\text{RM}_k$ | | | $\text{SM}_k$ | | | $\text{SJLT}_k$ | | | $\text{SJLT}_k \circ \text{RM}_{4k_{max}}$ | | | $\text{SJLT}_k \circ \text{SM}_{4k_{max}}$ | | | $\text{FJLT}_k$ | | |
| $k$ | 2048 | 4096 | 8192 | 2048 | 4096 | 8192 | 2048 | 4096 | 8192 | 2048 | 4096 | 8192 | 2048 | 4096 | 8192 | 2048 | 4096 | 8192 |
|---|---|---|---|---|---|---|---|---|---|---|---|---|---|---|---|---|---|---|
| LDS | 0.3690 | 0.4116 | 0.4236 | 0.3709 | 0.4020 | 0.4292 | 0.4131 | **0.4499** | 0.4747 | 0.4123 | 0.4357 | 0.4545 | 0.4104 | 0.4374 | 0.4581 | **0.4157** | 0.4497 | **0.4753** |
| Time (s) | **0.1026** | 0.1074 | 0.1296 | 0.1032 | **0.0879** | **0.1134** | 12.3590 | 12.2393 | 17.4836 | 0.3652 | 0.3648 | 0.3993 | 0.3054 | 0.2954 | 0.2911 | 31.5491 | 48.1669 | 81.9322 |

(c) LDS and compression wall-time for MusicTransformer with MAESTRO on TRAK.

|  | Sparsification | | | | | | Sparse Projection | | | GRASS | | | | | | Baseline | | |
|  | $\text{RM}_k$ | | | $\text{SM}_k$ | | | $\text{SJLT}_k$ | | | $\text{SJLT}_k \circ \text{RM}_{4k_{max}}$ | | | $\text{SJLT}_k \circ \text{SM}_{4k_{max}}$ | | | $\text{FJLT}_k$ | | |
| $k$ | 2048 | 4096 | 8192 | 2048 | 4096 | 8192 | 2048 | 4096 | 8192 | 2048 | 4096 | 8192 | 2048 | 4096 | 8192 | 2048 | 4096 | 8192 |
|---|---|---|---|---|---|---|---|---|---|---|---|---|---|---|---|---|---|---|
| LDS | 0.2773 | 0.2857 | 0.3194 | 0.2662 | 0.3273 | 0.3733 | **0.3062** | 0.3533 | 0.3861 | 0.2826 | 0.3378 | 0.3755 | 0.2539 | 0.3283 | 0.3657 | 0.2907 | **0.3585** | **0.4011** |
| Time (s) | 0.5341 | 0.5067 | 0.5179 | **0.3800** | **0.3971** | **0.4345** | 21.6460 | 21.1881 | 21.3192 | 0.7620 | 0.7532 | 0.7433 | 0.7487 | 0.7507 | 0.7495 | 100.8136 | 156.0613 | 269.9093 |

(d) LDS and compression wall-time for GPT2-small with WikiText on (block-diagonal FIM) influence function.

|  | Sparsification | | | | | | Sparse Projection | | | FACTGRASS | | | | | | LOGRA (Baseline) | | |
|  | $\text{RM}_{k_l^{in}\otimes k_l^{out}}$ | | | $\text{SM}_{k_l^{in}\otimes k_l^{out}}$ | | | $\text{SJLT}_{k_l^{in}\otimes k_l^{out}}$ | | | $\text{SJLT}_{k_l} \circ \text{RM}_{2k_l^{in}\otimes 2k_l^{out}}$ | | | $\text{SJLT}_{k_l} \circ \text{SM}_{2k_l^{in}\otimes 2k_l^{out}}$ | | | $\text{GAUSS}_{k_l^{in}\otimes k_l^{out}}$ | | |
| $k_l$ | 256 | 1024 | 4096 | 256 | 1024 | 4096 | 256 | 1024 | 4096 | 256 | 1024 | 4096 | 256 | 1024 | 4096 | 256 | 1024 | 4096 |
|---|---|---|---|---|---|---|---|---|---|---|---|---|---|---|---|---|---|---|
| LDS | 0.1034 | 0.1479 | **0.2391** | 0.0997 | 0.1617 | 0.2267 | **0.1240** | **0.1897** | 0.2389 | 0.1126 | 0.1784 | 0.2360 | 0.1102 | 0.1860 | 0.2380 | 0.1188 | 0.1818 | 0.2338 |
| Time (s) | **5.4933** | **5.3643** | **5.6385** | 5.8603 | 6.0436 | 5.8272 | 132.5404 | 133.4029 | 136.5163 | 6.5790 | 7.4161 | 6.3075 | 7.3443 | 8.5750 | 6.3330 | 20.4839 | 20.9835 | 22.2157 |

[4]We omit $\text{GAUSS}_k$ since the projection matrices for these two models are too large to fit in the GPU memory.

## 4.2 Scaling up to billion-size model

To evaluate the practical utility of FACTGRASS in attributing billion-scale models and datasets, we consider Llama-3.1-8B-Instruct [Meta AI, 2024] with a random 1B-token subset of the OpenWebText dataset [Gokaslan et al., 2019], and apply FACTGRASS (specifically, $\text{SJLT}_{k_l} \circ \text{RM}_{2k_l^{\text{in}} \otimes 2k_l^{\text{out}}}$) with layer-wise block-diagonal FIM influence function for linear layers. The experiment is conducted with one NVIDIA H200 GPU with 96 GB of memory, and more details can be found in Section B.3.

**Efficiency.** We measure the efficiency through the throughputs for FACTGRASS and LOGRA. Table 2 shows the throughput of 1.) **compress steps**: compute the projected gradients from inputs and gradients of pre-activation, and the overall 2.) **cache stage**: compute and save the projected gradients.

We see that FACTGRASS significantly improves compute efficiency compared to the previous SOTA, LOGRA. In terms of compression steps, we achieve a 160% faster throughput

Table 2: Throughput (tokens per second) for Llama-3.1-8B-Instruct.

|  | **Compress** | | | **Cache** | | |
|---|---|---|---|---|---|---|
| $k_l$ | 256 | 1024 | 4096 | 256 | 1024 | 4096 |
| LOGRA | 27,292 | 27,255 | 26,863 | 7,307 | 7,478 | 7,367 |
| FACTGRASS | **72,218** | **72,684** | **73,811** | **8,584** | **8,594** | **8,681** |

compared to LOGRA, which subsequently improves the overall caching throughput by around 17%. We note that the memory usages are similar in both cases: we set the batch to be 7 that maximizes the usage of memory bandwidth for both LOGRA and FACTGRASS.

**Qualitative Accuracy.** We next assess the qualitative alignment between the outputs generated by LLMs and the influential data identified by FACTGRASS with $k_l = 4096$. Since naive influence functions often highlight outlier data (e.g., error messages, ASCII codes, or repetitive words) with disproportionately high gradient norms [Choe et al., 2024], we filter these cases and select the most contextually relevant samples from the top-10 influential data identified by FACTGRASS. A representative example is presented in Figure 9. Given the simple prompt, "To improve data privacy," FACTGRASS identifies a paragraph discussing journalist jailings, including references to privacy policies on various news websites. This content closely aligns with the generated outputs from the model, demonstrating the qualitative accuracy of FACTGRASS in capturing relevant data influences.

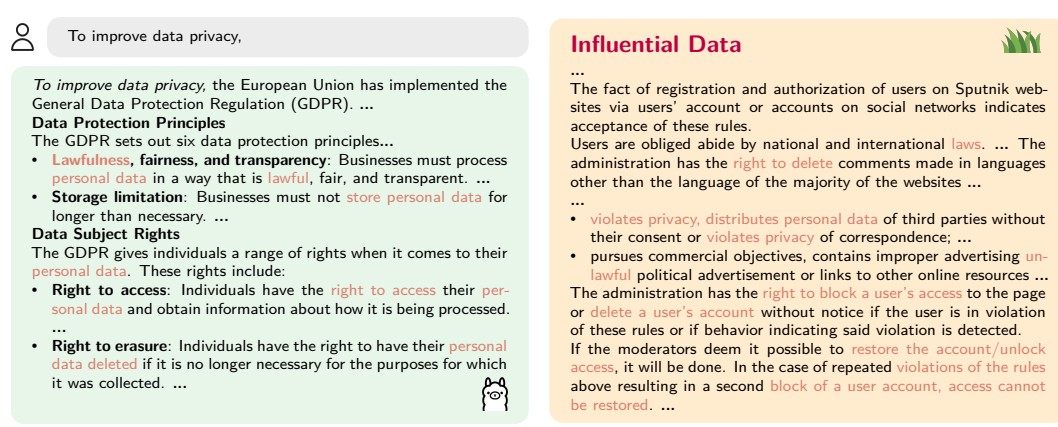

Figure 9: Qualitative accuracy of data attribution with FACTGRASS on Llama-3.1-8B-Instruct.

## 5 Conclusion

In this paper, we proposed GRASS, a novel gradient compression algorithm that leverages the inherent sparsity of per-sample gradients to reduce memory and computational overhead significantly. Building on this, we introduced FACTGRASS, a variant GRASS that further exploits the gradient structure of linear layers, achieving substantial practical speedups by avoiding ever materializing the full gradient that is both theoretically and practically faster than the previous SOTA baselines.

Our extensive experiments demonstrate that GRASS and FACTGRASS consistently outperform existing approaches in both efficiency and scalability, particularly on billion-scale language models.

## Acknowledgment

This work is supported in part by a NAIRR Pilot grant NAIRR240134. WT is partially supported by NSF DMS grant No. 2412853, and HZ was partially supported by an NSF IIS grant No. 2416897, an NSF CAREER Award No. 2442290 and a Google Research Scholar Award. We also thank Xueshen Liu from the University of Michigan for his invaluable discussion in improving the efficiency of our library's practical implementation. The views and conclusions expressed herein are those of the authors and do not necessarily reflect the official policies or positions of the supporting companies or government agencies.

## Broader impacts

This paper presents work whose goal is to advance the efficiency of data attribution algorithms. As there are many socially important applications of data attribution, with the fact that the present SOTA reliable data attribution methods do not scale well to commercial-size LLMs, there are many potential societal consequences of our work. However, none of which we feel must be specifically highlighted here.

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

# A  Omitted details from Section 2

## A.1  Related work

This section provides an in-depth discussion of gradient-based data attribution methods and their computational advancements, offering a holistic view of the field. For a more complete overview of other data attribution methods and their applications, we refer interested readers to the recent survey [Deng et al., 2025].

### A.1.1  Gradient-based data attribution

Gradient-based data attribution methods represent a major category for quantifying the impact of individual training samples on a model's behavior [Deng et al., 2025]. These techniques operate under the principle of using the model's local sensitivity—captured by the *per-sample gradient* with respect to model parameters—to infer how a training example contributes to model predictions or loss values.

The most influential and widely studied approach in this category is the *influence function* [Koh and Liang, 2017]. Rooted in robust statistics, the influence function provides a computationally efficient, first-order approximation of the expensive leave-one-out (LOO) retraining scenario. The core idea is to estimate the change in a target function (such as the loss on a test point) by calculating how the removal of a training sample perturbs the final model parameters. This estimation involves the per-sample gradient and the inverse Hessian matrix. Recent advancements, such as TRAK [Park et al., 2023], have been developed to improve on this classical framework.

Another family of gradient-based techniques called *training dynamic methods* has become popular due to its ability to adapt to complex training scenarios that might break the influence function assumptions. These include TracIn [Pruthi et al., 2020], SGD-Influence [Hara et al., 2019], and, more recently, data value embedding [Wang et al., 2025]. They analyze how training samples dynamically affect model behavior across intermediate checkpoints of the training trajectory, making them well-suited for non-convex settings where the classical influence function approximations may be less accurate.

All gradient-based methods ultimately rely on similar low-level computations—most notably the inverse Hessian–vector product (iHVP)—which makes the influence function a representative example for discussion.

### A.1.2  Scaling up gradient-based data attribution

The literature on scaling up gradient-based data attribution methods, or more specifically, iHVP computation, is extensive. In Section 2, we provided a detailed discussion of the RANDOM method, including various random projection techniques. Additionally, in Section 3.3.2, we briefly covered block-diagonal, layer-wise independence approximations of the Fisher information matrix (FIM), which together represent the most popular state-of-the-art in efficient approximations of iHVP. Here, we expand on this discussion by providing additional pointers to the broader literature on random projection and sketching for gradient compression, as well as alternative approaches to scaling up iHVP. This context helps clarify the positioning of our proposed methods within the larger landscape of scalable gradient-based data attribution research, offering a more complete understanding of the field.

**Sketching and RANDOM.**  Random projection, or sketching, is a well-studied technique for dimensionality reduction, widely explored in both theoretical computer science [Woodruff et al., 2014, Mahoney, 2016] and machine learning [Zhang et al., 2018, Liu et al., 2021, Chen et al., 2019, Li and Li, 2023]. In the context of gradient compression, sketching plays a critical role in distributed training, where the overhead of communicating full gradients can be a major bottleneck [Lin et al., 2018, Aji and Heafield, 2017]. However, direct sketching is often avoided in this context, as it can destroy important gradient information for the exact parameter correspondence of the gradient. Instead, techniques like random dropout, which selectively transmit parts of the gradient while preserving critical information, are more common. This is closely related to the Random Mask approach discussed in Section 3.2.

With the rise of gradient-based attribution methods, gradient compression has also become relevant for data attribution [Schioppa, 2024, Lin et al., 2024, Choe et al., 2024]. Among these, Schioppa

[2024] approaches gradient compression from a theoretical sketching perspective, refining traditional methods like the fast Johnson-Lindenstrauss transform (FJLT) [Ailon and Chazelle, 2009, Fandina et al., 2023] to improve computational efficiency on modern machine learning hardware such as TPUs. Notably, only Choe et al. [2024] specifically considers the structural properties of gradients when designing compression methods, potentially offering more accurate reconstructions with reduced communication cost.

**Input-Output Independence.** Two notable extensions of the block-diagonal approximation for empirical Hessians have emerged recently and can be further integrated with RANDOM. The first, Kronecker-Factored Approximate Curvature (K-FAC) [Martens and Grosse, 2015], leverages the Kronecker-factor structure of linear layers (same as Eq. (2)) by assuming independence between the inputs and the pre-activation gradients. This independent factorization significantly reduces the computational burden of Hessian approximations, as inverse FIM-vector product (iFVP) now only requires two smaller inversions of the factorized matrices. Compared to LOGRA, where the projected gradients and FIM are materialized at the end with the FIM of size $k^2$, K-FAC now only materializes two smaller projected inputs and gradients of the pre-activations and their corresponding covariances. The latter two covariances are of size roughly $\sqrt{k} \times \sqrt{k}$, further reducing the matrix inversion computation.

Building on this, Eigenvalue-corrected K-FAC (EK-FAC) [Grosse et al., 2023] refines this approach by correcting the eigenvalues of the factorized covariances, improving approximation quality without compromising efficiency. However, we note that while EK-FAC enhances the accuracy of K-FAC, it does not offer further computational speedups.

**Direct iHVP.** An alternative approach to scaling influence functions involves directly estimating the iHVP without explicitly forming or inverting the full Hessian. Unlike the two-stage methods discussed in Section 2, which approximate the full Hessian first and then compute iHVP for each training sample, this direct approach aims to bypass the costly inversion step, providing a more scalable solution for large-scale models.

One such algorithm is LiSSA [Agarwal et al., 2017], initially developed for stochastic optimization and later adapted for influence function calculations [Koh and Liang, 2017]. It approximates iHVP through iterative stochastic updates that *only involve* Hessian-vector product, which is efficient to compute. While straightforward, this method requires careful tuning of its hyperparameters to balance accuracy and runtime.

More recently, DataInf [Kwon et al., 2024] introduced a less conventional approach by reordering the sequence of matrix operations in the iHVP calculation. This method effectively swaps the expectation and inversion steps, allowing per-sample gradient information to approximate the inverse directly. However, this strategy tightly couples the iHVP estimation with the influence calculation, making it challenging to efficiently scale to large datasets, as it requires full computation for each training and test sample pair.

### A.2 A note on Johnson-Lindenstrauss lemma

While the Johnson-Lindenstrauss lemma [Johnson, 1984] ensures that the pair-wise distance, and hence the inner products, between gradients are approximately preserved under random i.i.d. projection, if $P$ is not injective when restricted to the range of $F_{\hat{\theta}}$, i.e., the column rank is not kept, then $F_{\hat{\theta}}$ applied to vectors in this subspace can generate significant components orthogonal to the subspace. In this case, the projected FIM $PF_{\hat{\theta}}P^\top$ implicitly neglects these orthogonal components, which leads directly to approximation errors [Schioppa et al., 2022]. Several potential strategies to mitigate this issue have been explored, focusing on how to construct the projection matrix $P$. One popular strategy is to first approximate the top-$k$ eigenvectors of $F_{\hat{\theta}}$ using classical algorithms such as PCA [Choe et al., 2024] and Arnoldi iteration [Schioppa et al., 2022, Arnoldi, 1951], then used the found top-$k$ eigenvectors as the rows of $P$.

## B Omitted details from Section 4

In this section, we provide further experimental details that we omitted in Section 4.

## B.1 Details of models, datasets, and computing resources

We summarize all the models and datasets used in the experiments in Table 3.

Table 3: Model details used in the experiments.

| Models | Datasets (License) | Task | Parameter Size | Train Samples | Test Samples | Sequential Length |
|---|---|---|---|---|---|---|
| MLP | MNIST (CC BY-SA 3.0) | Image Classification | 0.11M | 5,000 | 500 | 1 |
| ResNet9 | CIFAR2 (MIT) | Image Classification | 4.83M | 5,000 | 500 | 1 |
| Music Transformer | MAESTRO (CC BY-NC-SA 4.0) | Music Generation | 13.3M | 5,000 | 178 | 1 |
| GPT2-small | WikiText (CC BY-SA 3.0) | Text Generation | 124M | 4,656 | 481 | 512 |
| Llama-3.1-8B-Instruct | OpenWebText (CC0-1.0) | Text Generation | 8B | 976,562 | NA | 1024 |

All the experiments in quantitative analysis are conducted on `Intel(R) Xeon(R) Gold 6338 CPU @ 2.00GHz` with a single `Nvidia A40 GPU` with 48 GB memory. On the other hand, the qualitative analysis experiment is conducted on the `VISTA`[5] cluster with one Grace Hopper (GH) node, where each GH node has one `H200 GPU` with 96 GB of HBM3 memory and one `Grace CPU` with 116 GB of LPDDR memory.

## B.2 Details of quantitative analysis

**Model Training.** For MLP, ResNet9, and Music Transformer, we utilize pretrained models from the `dattri` library [Deng et al., 2024a, Appendix C]. For GPT2-small, we fine-tune the model on the WikiText dataset using the AdamW optimizer [Loshchilov and Hutter, 2019] with a learning rate of $5 \times 10^{-5}$ and no weight decay, training for 3 epochs.

**Linear Datamodeling Score (LDS).** We measure LDS using 50 data subsets, each containing half of the original training set. For each subset, models are trained independently using the hyperparameters described above. For a more comprehensive explanation of the LDS evaluation, we refer readers to Park et al. [2023].

**Data Attribution.** For MLP on MNIST, ResNet9 on CIFAR2, and Music Transformer on MAE-STRO, we use TRAK [Park et al., 2023] with 10, 10, and 5 independently trained checkpoints, respectively, as the backbone data attribution algorithm to evaluate different gradient compression methods. For GPT2-small fine-tuned on WikiText, we employ a layer-wise block-diagonal FIM approximation for linear layers as the backbone data attribution method.

We remark that one of the important hyperparameters that requires careful attention is the damping term used for the Hessian/FIM inverse. We pick the damping $\lambda$ for each setting (each model/dataset/compression method combination) via cross-validation grid search for LDS over $\lambda \in \{10^{-7}, 10^{-6}, 10^{-5}, 10^{-4}, 10^{-3}, 10^{-2}, 10^{-1}, 1, 10, 10^2\}$ on 10% of the test dataset, and evaluate the overall LDS result on the remaining 90% of the test dataset.

## B.3 Details of qualitative analysis

**Model and Dataset.** For Llama-3.1-8B-Instruct, we directly load the pretrained model without fine-tuning. As for the attribution dataset, while we do not have access to the massive 15T-token pre-training dataset used by Llama-3.1-8B-Instruct, we anticipate that it will contain most of the OpenWebText dataset due to its high quality and popularity.

## B.4 Practical implementation

Finally, we provide some remarks on the practical implementation of our proposed algorithms.

### B.4.1 SJLT implementation

A naive implementation of SJLT is straightforward: we first sample random indices corresponding to each input dimension along with their associated signs, and then perform a `torch.Tensor.index_add_()` operation, which carries out the core computation of SJLT (Figure 2). This operation implicitly uses atomic addition, which can lead to race conditions and slow

---

[5]See https://docs.tacc.utexas.edu/hpc/vista/.

down computation when the parallelization is not done carefully and the target compression dimension $k$ is small relative to the input dimension $p$.[6]

In contrast, our CUDA kernel implementation adopts a key optimization: we parallelize the computation by dividing the input dimension across different threads. This strategy reduces race conditions caused by atomic additions at each step.

### B.4.2 Selective Mask

We discuss several practical considerations and tips for solving Eq. (1) in the context of Selective Mask.

**Ensuring Exact $k$.** Since the sparsity of the mask arises from $\ell_1$ regularization, it is generally not possible to guarantee that the final $S^*$ contains exactly $k$ active indices (i.e., entries greater than $0.5$). A simple workaround is to select the top-$k$ indices based on their sigmoid values—i.e., adaptively setting the activation threshold to ensure exactly $k$ active indices. However, this method may yield suboptimal masks if the resulting values are far from binary, potentially degrading performance.

To address this issue, we increase the regularization strength and introduce an inverse-temperature term by replacing $S$ with $S/T$, where $T$ decreases as training progresses. As $T$ approaches zero, $\sigma(S/T)$ becomes more binary-like, promoting a "hard" mask. That said, empirical results show that careful tuning of the regularization parameter $\lambda$, combined with top-$k$ selection, can yield performance comparable to the inverse-temperature approach.

**Linear Layer.** For linear layers, we can derive a *factorized* Selective Mask by decomposing the gradients according to Eq. (2). Specifically, following the notation established in Section 3.3.2, for the $l^{\text{th}}$ linear layer, we can reformulate Eq. (1) as:

$$\underset{\substack{S_l^{\text{in}} \in \mathbb{R}^{d_l^{\text{in}}} \\ S_l^{\text{out}} \in \mathbb{R}^{d_l^{\text{out}}}}}{\arg\max} \mathbb{E}_{z_{\text{test}}} \left[ \text{corr}\left( (\langle z_{i,l}^{\text{in}} \otimes z_{i,l}^{\text{out}}, z_{\text{test},l}^{\text{in}} \otimes z_{\text{test},l}^{\text{out}} \rangle)_{i=1}^n, (\langle \hat{z}_{i,l}^{\text{in}} \otimes \hat{z}_{i,l}^{\text{out}}, \hat{z}_{\text{test},l}^{\text{in}} \otimes \hat{z}_{\text{test},l}^{\text{out}} \rangle)_{i=1}^n \right) \right]$$

$$- \lambda(\|\sigma(S_l^{\text{in}})\|_1 + \|\sigma(S_l^{\text{out}})\|_1),$$

where $z_{i,l}^{\text{in}} \in \mathbb{R}^{d_l^{\text{in}}}$ denotes the input feature of the $l^{\text{th}}$ linear layer from $z_i$, $z_{i,l}^{\text{out}} \in \mathbb{R}^{d_l^{\text{out}}}$ denotes the gradient of its pre-activation of the $l^{\text{th}}$ linear layer, and $\hat{z}_{i,l}^{\text{in}} = \sigma(S_l^{\text{in}}) \odot z_{i,l}^{\text{in}}$, $\hat{z}_{i,l}^{\text{out}} = \sigma(S_l^{\text{out}}) \odot z_{i,l}^{\text{out}}$ are the (soft-)masked variants. Computationally, we leverage the Kronecker product structure to simplify inner product calculations; for example:

$$\langle z_{i,l}^{\text{in}} \otimes z_{i,l}^{\text{out}}, z_{\text{test},l}^{\text{in}} \otimes z_{\text{test},l}^{\text{out}} \rangle = \langle z_{i,l}^{\text{in}}, z_{\text{test},l}^{\text{in}} \rangle \cdot \langle z_{i,l}^{\text{out}}, z_{\text{test},l}^{\text{out}} \rangle.$$

As a result, training with the Selective Mask does not require computing full layer-wise gradients, providing similar computational and memory efficiency as in FACTGRASS.

---

[6] A significant slowdown was previously observed when $k$ was extremely small, e.g., $k = 32$. However, with recent updates to `PyTorch`, this issue has been resolved, leading to consistent runtimes across different values of $k$, as shown in Figure 4.

