# OpenReview forum: "GraSS: Scalable Data Attribution with Gradient Sparsification and Sparse Projection"
_NeurIPS.cc/2025/Conference — NeurIPS 2025 poster_

### Official Review · Reviewer_iWVy · 2025-06-30

**Clarity:** 3
**Significance:** 3
**Originality:** 2
**Rating:** 4
**Confidence:** 5

**Summary:**

This paper introduces GraSS , a new method for making gradient-based data attribution more efficient. The key idea in this work is to exploit the sparsity of these per-sample gradients, most of the values are close to zero, and use that to compress the gradients. They also introduced a specialized version called FactGraSS , designed specifically for linear layers, which gives even better performance. In experiments, their method was significantly faster up to 165% faster throughput on huge models while still keeping the accuracy of influence estimation.

**Questions:**

1. Most current attribution methods, especially those based on Leave-One-Out (LOO) influence analysis, are built upon the Influence Function introduced by Koh and Liang. The core computation involves the product of the inverse Hessian matrix and the gradient. This framework has two main issues:

**Efficiency** : Computing or approximating the inverse Hessian is time-consuming and difficult, which has led to many follow-up works focusing on how to accelerate this part.

**Accuracy** : Influence Function-based methods assume the loss function is strongly convex, which theoretically limits their maximum achievable accuracy.

Later, more accurate methods emerged, such as SGD-Inf. My question is: Can newer acceleration techniques like GraSS also be applied to speed up these more advanced attribution methods?

---

2. Building on the first question, if the issue of speed is largely resolved, does the accuracy of current attribution methods still fall significantly short of what is required in real-world applications? Take SGD-Inf, one of the most accurate methods available today, as an example: its error grows exponentially with the number of training steps (exp(T)), meaning the error deteriorates rapidly as training progresses.

---

3. Is such a level of error still acceptable in practical scenarios? The third question again concerns the computational bottleneck. Even with efficient algorithms like GraSS or DataInf, the overall computation time for attribution still requires at least one full pass over the training data — that is, it has O(N) complexity. Meanwhile, in many large-scale model trainings today, the model is often trained for only one or a few epochs. This implies that the cost of attribution may already approach or even exceed the cost of the entire training process.
In such cases, how viable is the application of data attribution methods in practice?

**Ethical Concerns:**

["NO or VERY MINOR ethics concerns only"]

**Final Justification:**

I keep my initial rate for this paper.

**Limitations:**

see Questions

**Paper Formatting Concerns:**

Not any.

**Quality:**

3

**Strengths And Weaknesses:**

strength:

The method demonstrates a significant improvement in computational efficiency.

Weaknesses:

The presentation of the paper is somewhat difficult to follow.

---

> ### Author Rebuttal · Authors · 2025-07-29
>
> We sincerely thank Reviewer iWVy for taking the time to review our paper and for their constructive feedback and recognition of our proposed algorithm and its potential impact. Below, we provide point-to-point responses to the questions and concerns raised.
> > The presentation is difficult to follow
>
> We appreciate the reviewer's feedback regarding the paper's presentation and acknowledge that some paragraphs could be clearer. We are committed to improving readability and flow.
>
> We will thoroughly revise the writing in the next version to enhance clarity and ensure the paper is easier to follow. We welcome any further specific suggestions the reviewer might have to help us achieve this goal.
>
> > Can GraSS/FactGraSS also be applied to speed up advanced attribution methods such as SGD-Inf?
>
> Yes, our proposed GraSS and FactGraSS gradient compression algorithms are general and broadly applicable to various data attribution methods that rely on per-sample gradients [1, 2, 3, 4].
>
> For instance, recent advancements like [1,4] (which are conceptually similar to SGD-Inf) utilize techniques akin to those in LoGra [2]. Our methods can be seamlessly integrated into such frameworks to achieve significant acceleration.
>
> For clarity of presentation within this paper, we focused our experimental scope primarily on the Influence Function and its variants TRAK [3]. However, integrating GraSS/FactGraSS with other, more recent data attribution methods like SGD-Inf and works such as [1, 4] is a promising application of our proposed methods.
>
> > Does the accuracy of current attribution methods still fall short of what is required in real-world applications?
>
> While we agree that the required accuracy for data attribution methods is application-specific, however, we would like to highlight that existing methods already demonstrate practical utility in many scenarios.
>
> Our work's core contribution is to significantly accelerate gradient-based data attribution, achieving state-of-the-art efficiency for well-known methods like Influence Function and TRAK. By resolving the critical computational bottleneck, we enhance the practical applicability of these valuable tools, making them feasible for larger scales. Our focus is on computational efficiency, an orthogonal problem to inherent algorithmic accuracy limitations.
>
> > Error and computational bottleneck remain
>
> We appreciate the reviewer's insightful questions regarding the practical viability of data attribution methods. Regarding the level of error, whether it is acceptable remains application-specific. As discussed above, current data attribution methods have already demonstrated their utility and found practical use in various real-world scenarios. Our work focuses on addressing the computational bottleneck to expand their applicability.
>
> Furthermore, we acknowledge that current gradient-based attribution methods inherently require $N$ per-sample gradients, implying a complexity comparable to at least one training epoch. This is widely considered a hard computational lower bound in the field, with recent advancements [1, 2] that achieve attribution within "one training run" being recognized as significant breakthroughs.
>
> While we agree that for the largest frontier models during their pre-training phase, an attribution cost approaching or exceeding a single training epoch might be prohibitive, this is not universally true. For fine-tuning phases or smaller models, this level of computational expense is often affordable and efficient enough for practical application. Our methods significantly enhance this affordability for the scenarios where it is viable.
>
> **References**:
>
> [1]: Data Shapley in One Training Run
>
> [2]: What is Your Data Worth to GPT? LLM-Scale Data Valuation with Influence Functions
>
> [3]: TRAK: Attributing Model Behavior at Scale
>
> [4]: Capturing the Temporal Dependence of Training Data Influence

---

> ### Author Response · Authors · 2025-08-01
>
> We thank the reviewer for acknowledging our rebuttal. We hope our rebuttal has well addressed the reviewer’s concerns about our paper, and we are more than happy to address any follow-up comments if there is any. If not, we hope the reviewer can consider raising the rating. Thank you again for your time.

---

### Official Review · Reviewer_SuDq · 2025-07-02

**Clarity:** 3
**Significance:** 2
**Originality:** 2
**Rating:** 4
**Confidence:** 3

**Summary:**

This paper addresses the problem of scalable, gradient-based data attribution. Influence functions, commonly used for scoring the importance of samples in a dataset, require computing a (inverse) Hessian-gradient product, which is extremely costly for very large models. The authors propose two novel methods, GRASS and FACTGRASS. GRASS sparsifies the gradients using their proposed "selective mask" technique, after which they apply the sparse Johnson-Lindenstrauss transform. FACTGRASS enables the use of the compression technique via an interesting modification of the LoGRA technique applied to layerwise sparsified gradients of linear layers. The proposed data attribution techniques are evaluated with the LDS benchmark on million scale models and simple vision/music datasets, as well as on Llama3-8B models using the OpenWebText dataset, showing improved efficiency of computation over current methods.

**Questions:**

I have the following questions:

* How does the gradient sparsification impact the accuracy of the influence function based attribution? That is, given a dataset, can you discuss how the GRASS/FactGRASS influence functions compare with a full influence function?
* Can you provide a bound on the (expected) deviation from the "true" influence function score (as given in Koh and Liang's paper)? Can you quantify the manner in which gradient sparsity influences this?

**Ethical Concerns:**

["NO or VERY MINOR ethics concerns only"]

**Final Justification:**

I've raised my score to 4 (Borderline accept). While I still have some doubts about the significance of the work, the authors answered all my questions, particularly those pertaining to correctness.

**Limitations:**

I feel a key limitation to this work is the scope. The work, as presented, only proposes a method for efficiently computing influence functions. While the results appear to be technically correct, this work does not address any deeper question other than a simple improvement in cost. Moreover, there are no rigorous theoretical results - stated as Lemmas or Theorems - of interest from a statistical standpoint, though the computational complexity guarantees are nice to have.

**Paper Formatting Concerns:**

As best as I can see, there are no formatting concerns with this paper.

**Quality:**

2

**Strengths And Weaknesses:**

## Strengths

* The paper is generally quite well written
* The methods proposed address the real problem of the scalability of gradient-based data attribution, and the proposed method has a significantly lower wallclock time than the baselines
* The computational speedup of the GrASS method is shown rigorously in sections 3-4.

## Weaknesses

* Theoretical analysis could be written more clearly. The computational complexity resutls could be stated as formal results, at least in the appendix. Moreover, a table comparing computational costs of the different methods would have helped the exposition significantly/.
* The GRASS and FACTGRASS influence functions should be written as numbered equations for clarity.
* There is no discussion on the statistical properties of the GRASS estimator. Please refer to the "Questions" section as well.
* The quantitative analysis with the LLM is not very clear, and should be more clearly expressed.
* I feel the scope of this work is slightly limited - please refer to the "Limitations" section.

---

> ### Author Rebuttal · Authors · 2025-07-29
>
> We sincerely thank Reviewer SuDq for taking the time to review our paper and for their constructive feedback. Below, we provide point-to-point responses to the questions, hopefully addressing the concerns raised.
> > Better presentation of theoretical results
>
> We appreciate the reviewer's valuable suggestions for enhancing the clarity of our theoretical analysis and computational complexity results.
>
> We will update the paper accordingly, incorporating formal theorems to clearly state our method's computational complexity. Furthermore, we will include a comprehensive table comparing the computational costs of GraSS and FactGraSS, and other popular state-of-the-art methods, to significantly aid exposition. A sample table is provided below:
>
> | **Method**     | ***RM/SM*** | ***SJLT*** | ***GraSS*** | ***FactGraSS*** | **Gaussian** |      **FJLT**      |   **LoGra**    |
> | :------------- | :-------: | :------: | :-------: | :-----------: | :----------: | :----------------: | :------------: |
> | **Complexity** |  $O(k)$   |  $O(p)$  |  $O(k')$  | $O({k'}^{2})$ |   $O(pk)$    | $O((p + k)\log p)$ | $O(\sqrt{kp})$ |
>
> where $p$ denotes the model size, $k$ denotes the target compression dimension, and $k'$ is a parameter such that for GraSS, $k \leq k' \ll p$, and for FactGraSS, $\sqrt{k} \leq k' \ll \sqrt{p}$. We note that both LoGra and FactGraSS are specific for linear layers, and neither requires materializing the layer-wise gradients.
>
> Regarding the influence functions for GraSS and FactGraSS, we will ensure these are presented as numbered equations for improved clarity.
>
> > Discussion on the error between GraSS/FactGraSS compared to the full influence function
>
> We appreciate the reviewer's interest in the statistical properties of the proposed estimators. In the literature, state-of-the-art methods like LoGra [1] and TRAK [2] adopt random projection, often motivated by the approximation error guarantees provided by the Johnson-Lindenstrauss (JL) Lemma. Our approach adopts SJLT, which comes with JL Lemma-type guarantees as well [6]. However, a theoretical error bound w.r.t. the full influence function remains a significant open challenge in the current literature for efficient data attribution methods, including LoGra [1] and TRAK [2]. Arguably, most of the gradient-based efficient data attribution methods are driven by empirical development.
>
> Given this context, we believe our proposed methods are well-motivated. Our extensive empirical validation demonstrates competitive performance, highlighting the practical utility and reliability of GraSS/FactGraSS within the current research landscape. Further theoretical development in this area represents a promising direction for future work.
>
> We will include the above discussion in the next version to better motivate our methods.
>
> > Quantitative analysis with LLM is unclear
>
> We acknowledge the reviewer's feedback regarding the clarity of our quantitative analysis with LLMs:
> * Our experiments include results for GPT2-small (fine-tuned on Wikitext), presenting both LDS scores (in **Table 1**) and efficiency metrics. For Llama3-8B (evaluated on a subset of OpenWebText), we report throughput (in **Table 2**) as an efficiency measure.
> * Our experimental setup closely adheres to the methodology established by the recent state-of-the-art [1]. We will provide a more detailed and self-contained description of this setup in the updated version of the paper to enhance clarity.
>
> > Scope of the work is limited: only proposes a method for efficiently computing influence functions, without theoretical results
>
> We contend that efficiency is a core and pressing challenge for all recent state-of-the-art gradient-based data attribution methods [1, 3, 4], and our proposed GraSS and FactGraSS methods offer broadly applicable gradient compression techniques, which are crucial components in nearly every modern gradient-based attribution approach [1,2,3,4,5]. Thus, addressing this fundamental efficiency bottleneck significantly advances the practical applicability and scalability of a wide range of gradient-based data attribution algorithms.
>
> Regarding the lack of rigorous statistical theoretical results, we would like to clarify:
>
> 1. The key components of our proposed methods, particularly SJLT, come with strong theoretical justifications for their approximation quality [6], as discussed in **Lines 138**-**144**.
> 2. While a full statistical analysis of the end-to-end attribution process is indeed complex, as noted in our response to the second response (i.e., "Discussion on the error between GraSS/FactGraSS compared to the full influence function"), the broader development and evaluation of data attribution methods have been largely driven by extensive empirical research [1, 2, 3].
> 3. We have conducted comprehensive empirical evaluations on two of the most popular gradient-based data attribution methods, Influence Functions and TRAK [2], demonstrating the effectiveness and practical utility of our approach.
> Therefore, we believe that the proposed methods offer a substantial improvement in a critical area, supported by both component-wise theoretical guarantees and strong empirical evidence, aligning with the current trajectory of research in data attribution.
>
> **References**
>
> [1]: What is Your Data Worth to GPT? LLM-Scale Data Valuation with Influence Functions
>
> [2]: TRAK: Attributing Model Behavior at Scale
>
> [3]: Capturing the Temporal Dependence of Training Data Influence
>
> [4]: Data Shapley in One Training Run
>
> [5]: Scaling Up Influence Functions
>
> [6]: A Sparse Johnson--Lindenstrauss Transform

---

> ### Author Response · Authors · 2025-08-03
>
> We thank the reviewer for acknowledging our rebuttal. We hope our rebuttal has well addressed the reviewer’s concerns about our paper and that the reviewer can consider raising the rating. And we would appreciate further feedback from the reviewer on any remaining concerns. Thank you again for your time.

---

> > ### Comment · Reviewer_49cd · 2025-08-05
> >
> > Thanks the additional results from the author. I have raised my score.

---

> > > ### Author Response · Authors · 2025-08-05
> > >
> > > We thank Reviewer 49cd for acknowledging our response and updating the rating. We really appreciate it.
> > > > In the meantime, this might be the wrong thread you intended to reply to : )

---

> ### Comment · Reviewer_SuDq · 2025-08-06
> **Response to Author Rebuttal**
>
> I thank the author for their thoughtful response. In particular, I liked the table summarizing the complexities of the compared attribution methods, and their commitment to improving the presentation of their theoretical results.
>
> However, I'm still not convinced by the discussion regarding the theoretical analysis as presented. While I accept that other papers in this space do not provide rigorous justification, I would like to see, at the very least, some formal analysis of the impact of using the sparse JL transform vs the canonical JL transform. In particular, the role of sparsity should be formally discussed. For instance, $nnz(A)$, there is a maximum number of nonzero entries in the projection matrix for the JL bounds to hold for the SJLT (I believe that it is at least of the order $1/\varepsilon$). So even if the gradients are naturally sparse, it seems odd to me that you can choose $s=1$ (as noted on line 144).

---

> ### Author Response · Authors · 2025-08-06
>
> We thank the reviewer for the insightful follow-up, and we would love to discuss how SJLT affects the performance compared to canonical JLT. The short answer is that, SJLT enjoys *asymptotically* the same JL-type guarantees compared to, e.g., dense Gaussian projection. Furthermore, by letting $s$ as a tunable parameter, our proposed methods still outperform the baselines significantly, both in theory and in practice. Please see below for a detailed discussion.
>
> ---
>
> Recall that the canonical JL Lemma guarantees:
>
> > (Informal) For any $d$ and $0 < \epsilon < 1 / 2$, there exists a probability distribution on $k \times d$ real matrices $S \in \mathbb{R} ^{k \times d}$ with $k = \Theta (\epsilon ^{-2})$ such that "for any" $x \in \mathbb{R} ^d$, $\mathbb{P} _S ((1 - \epsilon ) \lVert x \rVert _2 \leq \lVert Sx \rVert _2 \leq (1 + \epsilon ) \lVert x \rVert _2) > 1 - \delta$. Note that we hide the dependency of $\delta$ in all bounds for simplicity.
>
> First, we want to clarify a potential confusion about the roles of “*per-sample gradient sparsity*” and “*projection matrix sparsity*” here: observe that the role of *per-sample gradient sparsity* (i.e., $\lVert x \rVert _0$) is **irrelevant** in terms of the JL-type guarantees, as the statement is for **any** vectors $x \in \mathbb{R} ^d$. Instead, the importance of the per-sample gradient sparsity only lies in the **computational aspect** of $Sx$: the sparser the per-sample gradient is, the faster we can obtain the projected per-sample gradient $Sx$.
>
> With this in mind, we next discuss $s$ for SJLT. [1] shows that there exists such a distribution over $S$ with $s = \Omega (\epsilon ^{-1})$, where $s$ is the number of non-zero entries of each column of $S$. We see that:
> 1. Choosing $s$ is **independent** of input sparsity.
> 2. As these bounds are asymptotic, they inform very little about how to choose $s$ in practice. With extensive empirical testing across different orders of $d$ and $k$, we find that in practice, $s = 1$ usually suffices and achieves relative errors similar to the fully dense Gaussian projection matrix (e.g., **Figure 2** in the paper).
>
> On the other hand, keeping $s$ as a tunable parameter will not change the competitiveness of our proposed method much:
>
> 1. Complexity of GraSS goes from $O(k’)$ to $O(sk’)$ for some $k \leq k’ \ll \sqrt{p}$ and $1 \leq s \ll k$.
> 2. Complexity of FactGraSS goes from $O({k’}^2)$ to $O((sk’)^2)$ for some $\sqrt{k} \leq k’ \ll \sqrt{p}$ and $1 \leq s \ll \sqrt{k}$.
>
> In both cases, our proposed methods outperform the respective SOTA baselines easily under any reasonable choice of $s$ in practice. We previously omitted this in our discussion due to the space limit. We will incorporate the above discussion into the next version.
>
> ---
>
> **References**:
>
> [1]: Sparser Johnson-Lindenstrauss Transforms

---

> > ### Comment · Reviewer_SuDq · 2025-08-07
> > **Response to reviewer comment**
> >
> > Thanks for the follow up. That has answered my questions, and I'll raise my score.

---

> > > ### Author Response · Authors · 2025-08-07
> > >
> > > We thank Reviewer SuDq for the constructive feedbacks and acknowledging our response. Thank you again for your time.

---

### Official Review · Reviewer_49cd · 2025-07-02

**Clarity:** 2
**Significance:** 2
**Originality:** 2
**Rating:** 4
**Confidence:** 4

**Summary:**

This paper proposes GRASS, a new gradient compression method to make influence functions faster and more memory-efficient for large neural networks with sparsification and projection.

**Questions:**

Q1. Can the authors provide a direct quantitative comparison between GRASS/FACTGRASS and RapidIn (Lin et al., 2024) on common benchmarks?

Q2. Can the authors add quantitative accuracy evaluation (such as LDS or other metrics) for the billion-scale model instead of only reporting speed and one qualitative example?

Q3. Can the authors analyze how the Selective Mask is chosen and whether its stability or importance affects the attribution results?

Q4. Can the authors include experiments or analysis showing how sensitive GRASS/FACTGRASS is to key hyperparameters like k and k'?

Q5. Can the authors explain or empirically study why random masking leads to lower accuracy and in what situations it is still a good choice?

**Ethical Concerns:**

["NO or VERY MINOR ethics concerns only"]

**Final Justification:**

The additional experiment shows the comparison with more baselines and I have raised the score.

**Limitations:**

yes

**Quality:**

2

**Strengths And Weaknesses:**

Strong Points
----
S1. The paper tackles the important problem of scaling influence functions to very large models.

S2. Combining gradient sparsity and sparse projection is a novel way to speed the influence function computation.

Weak Points
----
W1. The paper mentions RapidIn (Lin et al., 2024), which also uses random projection, but does not include a direct experimental comparison or discussion of pros and cons versus RapidIn.

W2. The experiments focus on only a few models, especially only one billion-scale model.

W3. The importance and stability of mask selection in Selective Mask is not analyzed much.

W4. The effect of different hyperparameter settings (like k and k') is not fully explored.

W5. Random masking loses accuracy compared to other methods.

W6. For the billion-scale model, the paper reports only speed and a single qualitative example, but does not provide any quantitative evaluation of data attribution accuracy.

---

> ### Author Rebuttal · Authors · 2025-07-29
>
> We sincerely thank Reviewer 49cd for taking the time to review our paper and for their constructive feedback. Below, we provide point-to-point responses to the questions, hopefully addressing the concerns raised.
>
> > Comparison with RapidIn.
>
> We appreciate the reviewer’s suggestion to include a more direct comparison with RapidIn. Here, we provide a detailed discussion and empirical comparison that we will also integrate into the next version.
>
>  **Theoretical Complexity**: In terms of computational complexity, the core RapidIn’s gradient compression method requires $O(\lambda p)$, where $\lambda$ is a hyperparameter that is of order $20$ to $100$, and $p$ is the model size. Compared to our proposed methods, which require only $O(k’)$ computational time for some $k < k’ \ll p$, we argue that our methods outperform RapidIn significantly.
>
> **Empirical Comparison**: As the actual runtime of each method highly depends on the specific implementation, it is tricky to compare different methods. Nevertheless, we run the official implementation of RapidIn to compare. Below are the results using RapidIn’s public codebase and default hyperparameters ($\lambda = 20$):
>
> *MLP+MNIST (RapidIn)*:
>
> * $k$=2048: LDS=0.1529, Time=4.5928
> * $k$=4096: LDS=0.1547, Time=4.6150
> * $k$=8192: LDS=0.1554, Time=4.7306
>
> *ResNet9+CIFAR2 (RapidIn)*:
>
> * k=2048: LDS=0.0234, Time=16.6967
> * k=4096: LDS=0.0235, Time=16.7008
> * k=8192: LDS=0.0223, Time=16.7235
>
> *MusicTransformer+MAESTRO (RapidIn)*:
>
> * $k$=2048: LDS=0.1268, Time=43.1692
> * $k$=4096: LDS=0.1338, Time=43.2024
> * $k$=8192: LDS=0.1432, Time=43.1993
>
> Compared to **Table 1** in our paper, our methods consistently achieve significantly lower projection times, often by several orders of magnitude, while yielding higher LDS values. We again note that the comparison has some caveats: first, our compression algorithms are implemented with a CUDA kernel, while theirs use only high-level PyTorch APIs; second, RapidIn is different from TRAK used in the paper: the former only requires one model checkpoint, while TRAK requires several (5~10) models ensemble, and the reported wall time is the total projection time for each attribution method. Besides all these caveats, we highlight that the *CUDA kernel implementation is also a key contribution of this work*, and it requires non-trivial efforts to implement a CUDA-kernel-equivalent algorithm for RapidIn.
>
> > Experiments focus on only a few models and only one billion-scale model.
>
> We respectfully clarify that our experimental setup includes **five** different models, each evaluated on a distinct dataset, covering a diverse range of architectures and domains. For the billion-scale setting, we adopt the largest experimental configuration used in LoGra [1], which serves as a strong benchmark for evaluating attribution methods at scale. We believe this setup sufficiently demonstrates the efficiency and scalability of our proposed methods relative to the current state-of-the-art.
>
> > Stability and implementation of Selective Mask.
>
> We address the two questions below, respectively:
>
> **Choosing Selective Mask**: The procedure for learning the Selective Mask is described in **Section 3.2.2**. Here, we provide additional implementation details and clarifications that were omitted from the main text due to space constraints:
> * *Forcing exact $k$*: A naive optimization for Selective Mask can not guarantee that the final $S^{\ast}$ contains exactly $k$ active indices. A simple workaround is to select the top-$k$ indices based on their sigmoid values, which corresponds to adaptively setting the activation threshold to ensure exactly $k$ active indices.
> * *Linear Layer*: For linear layers, we can derive a **factorized** Selective Mask by decomposing the mask w.r.t. the gradients factorization, according to **Eq. 2**. Specifically, one can reformulate **Eq. 1** and leverage the Kronecker product structure to simplify inner product calculations. As a result, training Selective Mask does not require computing full layer-wise gradients, providing similar computational and memory efficiency as in FactGraSS. This additional discussion will be included in the next version.
>
> **Stability**: We refer to the ablation studies in **Table 1** (specifically the standalone Selective Mask (SM) columns), where we evaluate SM across multiple settings and varying target projection dimensions. The results demonstrate that SM is stable across runs and consistently outperforms the baseline Random Mask in terms of LDS, highlighting its effectiveness and robustness. Furthermore, we conducted additional ablation studies to assess the sensitivity and impact of using the Selective Mask alone as a gradient compression method on attribution results. Below, we report the *mean* LDS and the *standard error of the mean* across five different random seeds:
>
> *MLP+MNIST (TRAK with SM)*:
>
> * $k$=2048: LDS=0.3924 (±0.0030)
> * $k$=4096: LDS=0.4192 (±0.0034)
> * $k$=8192: LDS=0.4274 (±0.0032)
>
> *ResNet9+CIFAR2 (TRAK with SM)*:
>
> * $k$=2048: LDS=0.3733 (±0.0071)
> * $k$=4096: LDS=0.4072 (±0.0028)
> * $k$=8192: LDS=0.4326 (±0.0037)
>
> *MusicTransformer+MAESTRO (TRAK with SM)*:
> * $k$=2048: LDS=0.1470 (±0.0030)
> * $k$=4096: LDS=0.1628 (±0.0039)
> * $k$=8192: LDS=0.1651 (±0.0033)
>
> This consistent LDS performance demonstrates the stability of the Selective Mask.
>
> > Add quantitative accuracy evaluation for billion-scale models beyond throughput.
>
> We agree that it would be valuable to include additional quantitative analysis beyond throughput. However, computing common metrics such as LDS at the 8B scale is prohibitively expensive in terms of time and memory. This limitation is well recognized in prior work [1], which similarly did not include such measurements at this scale due to the prohibitively high computational cost. Our experimental setup follows closely with [1], aligning with the literature.
>
> > Sensitivity of GraSS/FactGraSS to key hyperparameters like $k$ and $k’$.
>
> We have addressed the sensitivity of GraSS/FactGraSS to the hyperparameters $k$ and $k’$,  by the experiments in **Section 4** of our paper. In these experiments, we consider three different target compression dimensions across various settings, including billion-scale models and datasets. The consistent performance trend observed across these values demonstrates the robustness of our approach to reasonable variations of compression dimensions.  Additionally, as discussed in **Lines 271**-**272**, we don't include results for larger values of $k$ and $k′$ because increasing $k$ leads to a quadratic complexity blow-up. This would require storing and inverting a matrix that far exceeds our available GPU memory resources. This practical limitation prevents further exploration of such extreme ranges.
>
> In terms of other hyperparameters, we follow the same settings as in [3] to align with the literature.
>
> > Why does Random Mask lead to lower accuracy, and when is it effective?
>
> Random Masking, along with Selective Masking and SJLT, serves as a component within our proposed algorithm's pipeline. The experiments involving Random Masking are primarily designed as ablation studies to facilitate comparisons with other algorithmic components and our final proposed methods.
>
> On the other hand, while Random Masking may lead to lower accuracy compared to more sophisticated compression techniques, its key advantage lies in its exceptional efficiency. This makes it the most efficient compression method within the pipeline. Consequently, its inclusion provides a valuable option for scenarios where a trade-off favoring computational efficiency over maximal accuracy is acceptable or necessary.
>
> **References**
>
> [1]: What is Your Data Worth to GPT? LLM-Scale Data Valuation with Influence Functions
>
> [2]: TRAK: Attributing Model Behavior at Scale
>
> [3]: 𝚍𝚊𝚝𝚝𝚛𝚒 : A Library for Efficient Data Attribution

---

> ### Author Response · Authors · 2025-08-01
>
> We thank the reviewer for acknowledging our rebuttal. We hope our rebuttal has well addressed the reviewer’s concerns about our paper and that the reviewer can consider raising the rating. And we would appreciate further feedback from the reviewer on any remaining concerns. Thank you again for your time.

---

### Official Review · Reviewer_HgYa · 2025-07-06

**Clarity:** 4
**Significance:** 4
**Originality:** 4
**Rating:** 5
**Confidence:** 4

**Summary:**

The paper proposes to leverage the per-sample gradient sparsity inherit in neural networks with relu activations to speed up influence function computation.  In particular, the sparse JL lemma is leveraged to accelerate random projections to take advantage of the sparsity.  Furthermore, additional extensions are proposed: 1) the parameters are first further sparsified via a one-time optimization problem 2) extending Logra, the input activation and output gradients in linear layers are sparsified, and a similar approach is applied to their respective Kronecker product for computational speedup.  The resulting method is able to achieve up to 165% speedup over Logra on Llama-3.1-8b-instruct.

**Questions:**

- is it possible to perform some quantitative analysis of FactGrass for the 8B model beyond throughput?

**Ethical Concerns:**

["NO or VERY MINOR ethics concerns only"]

**Final Justification:**

The discussion did not lead to new points which would change my rating.

**Limitations:**

yes

**Quality:**

3

**Strengths And Weaknesses:**

Strengths
- the paper proposes a novel core idea - leveraging / introducing parameter sparsity to speedup influence function computations.  this idea could likely be extended to other problems involving per-sample gradient compression
- the experiments show show significant speedup over Logra on billion-parameter models

Weaknesses,
- analysis focuses only on 1 billion-parameter model, unlike Logra (which considered 3), though this is understandable given computational resources.

---

> ### Author Rebuttal · Authors · 2025-07-29
>
> We sincerely thank Reviewer HgYa for taking the time to review our paper and for their constructive feedback and recognition of our proposed algorithm and its potential impact. Below, we provide point-to-point responses to the questions and concerns raised.
> > Only 1 billion-scale model in empirical analysis.
>
> Due to computational constraints, we focused our quantitative analysis on smaller models and efficiency analysis on an 8B-parameter model. It is worth noting that our experimental setup follows closely to the previous state-of-the-art [1]. Specifically, for efficiency analysis, we highlight that:
> 1. The efficiency analysis is conducted on the 8B model, the largest model [1] considered.
> 2. The observed speedup is robust and model-agnostic, as supported both by our computational complexity analysis (independent of model) as well as the empirical evidence (up to overall 165% throughput speed up).
>
> Hence, we believe the quantitative analysis results, as well as efficiency results for 8B model, provide a representative and reliable demonstration of the efficacy.
>
> > Possibility to perform quantitative analysis for 8B model beyond throughput.
>
> We agree that it would be valuable to include additional quantitative analysis beyond throughput. However, computing common metrics such as LDS at the 8B scale is prohibitively expensive in terms of time and memory. This limitation is well recognized in prior work [1], which similarly did not include such measurements at this scale due to the prohibitively high computational cost.
>
> **Reference**
>
> [1]: What is Your Data Worth to GPT? LLM-Scale Data Valuation with Influence Functions

---

> > ### Comment · Reviewer_HgYa · 2025-08-08
> >
> > Thank you to the authors for the clarification regarding the quantitative analysis that is feasible and done by other papers.  Looking at the discussion, I maintain my original view that the paper, while somewhat narrow in scope - targeting an empirical improvement to a specific efficiency problem - presents a novel and potentially impactful approach, and thus keep my original score.

---

### Note · Authors · 2025-08-12

Dear Reviewers and ACs,

We would like to offer the following final remarks to provide a broad perspective on our work and to summarize the author–reviewer discussion.

In this work:

1. We propose **sub-linear** gradient compression algorithms with **efficient implementations**, achieving state-of-the-art computational efficiency.
2. Through extensive experiments, we demonstrate the algorithm’s effectiveness for influence function computation.

Reviewers have commonly acknowledged the following strengths:

1. A **novel algorithmic design** supported by rigorous computational complexity analysis.
2. Directly addressing the **real scalability challenge** in gradient-based attribution methods.
3. Significantly **reduced** wall-clock time compared to state-of-the-art baselines **in practice**.
4. Potential **applicability** of the technique to a **broader class of problems** involving per-sample gradient compression.

During the rebuttal phase, we addressed the following points:

1. Perceived limitation in scope
   * Our approach is **not** confined to influence functions; for example, it can be integrated with other data attribution methods leveraging per-sample gradients, opening promising directions and applications. More broadly, as acknowledged by reviewers, our methods can be potentially applied to broader class of problems that involve per-sample gradient compression.
2. Sensitivity to key hyperparameters
   * We have provided additional experiments confirming the statistical stability of our method.
3. Limited quantitative analysis beyond throughput for billion-scale experiments
   * Due to the intense computational requirement, we follow the largest experimental setting employed by prior state-of-the-art.
4. Comparison with other baselines
   * We demonstrated that our method is in practice several orders of magnitude faster than other baselines

We appreciate the constructive feedback provided during the review process. We believe the proposed method offers both theoretical novelty and practical value, addressing a key bottleneck in scalable data attribution. We hope our clarifications and additional results have addressed the remaining concerns, and we look forward to contributing this work to the community.

---

### Decision · Program_Chairs · 2025-09-17

**Decision:**

Accept (poster)

**Comment:**

The authors propose a method for efficient computation of gradient-based data attribution methods that exploits the natural sparse structure of per-sample gradients in ReLU networks and well-adapted randomized projection methods, in particular the sparse JL transform. The methodology carefully integrates different techniques from prior work with this core approach and different strategies for masking. The authors provide a detailed complexity analysis of their method, as well as CUDA kernels for key components. Experimental results demonstrate significant speedups on large models.

Initial reviews for the paper were positive, and they further increased after the rebuttal, in which the authors clarified the potential for their method to be applied beyond computation of influence functions and provided additional comparisons to baselines. Concerns that the results did not demonstrate advantages beyond 1B scale were seen as acceptable by the reviewers after discussion. The AC agrees with the reviewers' judgment, and recommends acceptance. The authors should carefully incorporate feedback from the reviewers into the final revision.